# LOCAL STEPSIZES ACCELERATE DISTRIBUTED OPTIMIZATION

## ABSTRACT

Distributed optimization is a core enabling technique for large-scale machine learning, multi-agent systems, and decentralized control, allowing both data and computation to be distributed across multiple agents. A key challenge in the design of distributed optimization algorithms lies in selecting appropriate step sizes. Most existing distributed algorithms rely on a coordinated global step size across the agents, which may be challenging to implement in a fully decentralized setting with many agents. Although some efforts have been made to develop adaptive or uncoordinated step size strategies for distributed optimization, these approaches generally exhibit inferior performance compared to coordinated methods, in which all agents use a common step size determined centrally or through a unified design procedure. In this work, we present a somewhat surprising finding that local step sizes for distributed optimization (with no coordination) can outperform their global step size counterparts. The results are obtained using a rigorous computer-assisted performance-characterizing technique (semidefinite programming) for optimization algorithms and are applicable to all convex and smooth objective functions. To the best of our knowledge, this is the first time that such results have been established for general objective functions in a rigorous and systematic manner. Experimental results using benchmark datasets confirm the theoretical discoveries.

## 1 INTRODUCTION

Distributed optimization serves a pivotal role in diverse domains, including multi-agent systems, wireless sensor networks, machine learning, and large-scale control systems, wherein both data and computational resources are inherently decentralized and dispersed across multiple agents (Tsitsiklis et al., 1986; Nedic and Ozdaglar, 2009b; Predd et al., 2009; Nedic et al., 2010; Nedić and Olshevsky, 2014; Koloskova et al., 2021). Most existing distributed optimization algorithms rely on a global universal step size for all agents (Shi et al., 2015; Nedic et al., 2017). However, determining such a global step size is inherently challenging, as it requires all agents to have knowledge of the global objective function information that is difficult to obtain in a decentralized setting. This is because acquiring such global knowledge necessitates substantial communication among agents, which can be costly in large-scale settings (Xu et al., 2015) or even impractical due to the privacy or legal concerns on inter-agent information sharing (Warnat-Herresthal et al., 2021). Additionally, using a global step size across all agents forces each agent to conform to the geometry of the global objective function, which can potentially lead to slow convergence. For example, using a special set of quadratic functions, Mukherjee et al. (2024) recently demonstrates that employing local step sizes that respect the local curvature of individual agents can accelerate convergence. However, a rigorous and systematic analysis of the potential benefits of local step sizes is still largely absent from the literature. It is worth noting that although distributed optimization results with uncoordinated step sizes have been explored in the control community (see, e.g., Xu et al. (2015; 2017); Nguyen et al. (2022); Nedić et al. (2017); Wang and Nedić (2023)), these results generally do not offer advantages in convergence speed compared with their coordinated counterparts, where a commensurate global step size is applied to all agents.

In this work, we leverage a powerful computational framework—the performance estimation problem (PEP) based on semidefinite programming (SDP) (Vandenberghe and Boyd, 1996; Taylor et al., 2016)—to rigorously characterize first-order optimization methods and systematically address the following question: *Can local step sizes respecting local geometries of individual objective functions*

*outperform their counterpart with a global universal step size in distributed optimization under general strongly convex objective functions?* To answer this question, we first generalize the existing PEP framework to a practical setting where the local optima of individual objective functions are always bounded. To this end, we incorporate additional constraint terms in the PEP formulation and prove that it can still ensure an **exact** characterization of the performance of a distributed optimization algorithm, i.e., (1) the obtained solution gives the worst-bound for all possible smooth strongly convex functions, and (2) at least one function exhibiting the corresponding performance bound are guaranteed to exist. Moreover, given that some distributed optimization algorithms are sensitive to the heterogeneity in step sizes, we focus on the NIDS algorithm (Li et al., 2019b) which has theoretically guaranteed convergence under local step sizes (it is worth noting that albeit having guaranteed convergence, the theoretical results in Li et al. (2019a) in fact imply that local step sizes reduce convergence speed compared with the case where a commensurate global step size is used by all agents).

Our results prove that local step sizes can accelerate distributed optimization compared with coordinated global step size counterparts, and the conclusion applies to general smooth and strongly convex objective functions, extending far beyond the example set of quadratic functions discussed in Mukherjee et al. (2024). (To ensure a fair comparison, the analysis was performed under the $\frac{1}{L}$ step size rule, where $L$ corresponds to the global Lipschitz constant in the global case and it is set to the local Lipschitz constant of each individual agent in the local case.) Building on this insight, we have developed a new algorithm to ensure faster convergence with local rather than global step sizes. This is highly nontrivial since many distributed optimization algorithms (including EXTRA (Shi et al., 2015), which inspired our algorithm) may diverge when heterogeneous step sizes are used.

**Contributions.** Our main contributions are summarized as follows:

- We rigorously prove that local step sizes can indeed provide faster convergence in distributed optimization than their global step size counterpart for general smooth and strongly convex objective functions. To our knowledge, such a result has only been reported for a very special set of quadratic functions with identical optimal solutions (see, e.g., Mukherjee et al. (2024)), and this is the first time that this result is established rigorously for a general class of objective functions.

- Inspired by this discovery, we have also provided a new distributed optimization algorithm that can indeed exploit local step sizes to accelerate distributed optimization. This algorithm is inspired by EXTRA (Shi et al., 2015) and NIDS (Li et al., 2019a) but has strictly faster convergence.

- To rigorously compare the performance of distributed optimization using local step sizes versus global coordinated step sizes, we generalized the existing distributed PEP framework in two key aspects: firstly, we modified it to accommodate the boundedness restriction of optimal solutions in practical applications of distributed optimization; secondly, we revised it to reduce computational complexity, which is crucial given that the existing PEP approach is computationally intensive. Our formulation applies to all convex or strongly convex functions that are smooth.

- Finally, we conduct a comprehensive set of experiments on both synthetic and real-world datasets to validate the correctness of our theoretical findings and demonstrate the effectiveness of the proposed algorithm. The synthetic experiments are designed to illustrate the effects of local step sizes under controlled conditions, allowing for clear comparisons between local and global step size strategies. The experiments using real-world datasets, on the other hand, demonstrate the practical applicability of our approach.

**Distributed Optimization** Distributed optimization has been extensively studied over the years (see Eckstein (1994); Nedic and Ozdaglar (2009b); Forero et al. (2010); Arjevani and Shamir (2015); Yang et al. (2019); Koloskova et al. (2021) for examples of some popular algorithms). Early research in this area primarily focused on optimization methods (see, e.g., Shi et al. (2015); Forero et al. (2010); Xi et al. (2017); Qu and Li (2017); Scaman et al. (2018); Xin and Khan (2018); Scaman et al. (2019); Qu and Li (2019); Pu et al. (2020)) that ensure convergence to the global optimal solution, despite each agent having access only to a local, partial view of the global objective function. In recent years, distributed optimization has garnered increased attention due to its applications in

collaborative machine learning (see, e.g., Haddadpour and Mahdavi (2019); Deng et al. (2020)). This has led to further developments in distributed learning techniques (see, e.g., Koloskova et al. (2019); Pu and Nedić (2021); Chen et al. (2021); Horváth et al. (2023)), as well as in learning techniques with new features such as privacy protection (see, e.g., Chen and Wang (2024b); Koloskova et al. (2023); Allouah et al. (2024)).

**Locally Adaptive Stepsizes**  Locally adaptive step sizes have gained increased attention in federated learning (distributed learning with a parameter server), with some typical examples including Mukherjee et al. (2024); Wang et al. (2021); Kim et al. (2023). However, most of these results do not discuss the comparison in convergence speed with respect to the case where a global coordinated step size is used. One exception is Mukherjee et al. (2024), which, using a special set of quadratic functions with equal optimal solutions, shows that local step sizes can yield faster convergence than the coordinated step size case. Adaptive step size has also been studied in the fully distributed setting (see, e.g., Chen et al. (2023); Chen and Wang (2024a)). However, none of these results show local adaptivity to be superior to the case where a global adaptive step size is used.

**Performance Estimation Problem**  The idea of obtaining exact performance measures for optimization algorithms—rather than relying on analytical, often conservative, asymptotic analyses—originated from Nesterov (2004) and was formalized by Drori and Teboulle (2014) through the Performance Estimation Problem (PEP) framework as follows:

$$\sup_{f, x_0, \ldots, x_K, x_*} \mathcal{P}(\mathcal{O}_f, x_0, \ldots, x_K, x_*) \tag{PEP}$$

$$\text{such that} \quad f \in \mathcal{F},$$
$$x^* \text{ is optimal for } f,$$
$$x_1, \ldots, x_K \text{ are generated from } x_0 \text{ by method } \mathcal{M} \text{ with } \mathcal{O}_f,$$
$$\|x_0 - x_*\|_2 \leq R,$$

where $\mathcal{M}$ denotes a fixed-step size optimization algorithm, $\mathcal{O}_f$ denotes the oracle that returns the corresponding function value and gradient at a given point of the function $f$, $\mathcal{F}$ is the class of functions to which $f$ belongs (e.g., strongly convex functions or general convex functions), $K$ is the number of steps the algorithm $\mathcal{M}$ performs, and $R$ is a given constant. By solving the PEP, one can obtain **exact** performance bounds for algorithms over designated function classes, in contrast to traditional asymptotic analyses, which often yield loose guarantees. Taylor et al. (2016); Taylor and Bach (2019) formulated the PEP as a convex semidefinite programming (SDP) (Vandenberghe and Boyd, 1996), which can be solved efficiently and yield **exact** performance bounds. Recently, this framework has been extended to distributed optimization (Colla and Hendrickx, 2022; 2023).

## 2 PROBLEM SETUP

### 2.1 DISTRIBUTED OPTIMIZATION SETUP

We consider the following distributed optimization problem across $N$ agents:

$$\min_{x \in \mathbb{R}^d} f(x) := \frac{1}{N} \sum_{i=1}^{N} f_i(x), \tag{1}$$

where $f_i : \mathbb{R}^d \to \mathbb{R}$ is private to agent $i$ for $i \in [N] := \{1, ...N\}$ and $\mathbb{R}^d$ denotes the underlying Euclidean space. Throughout this work, we assume that the global function $f(x)$ is bounded below by $f^* > -\infty$. To minimize $f(x)$ in a fully distributed setting, each agent $i$ holds its own local estimate $x_i$ of the decision variable $x \in \mathbb{R}^d$. Each agent performs local computations and exchanges information with neighboring agents to ensure consensus across all agents' solutions. The communication pattern can be formulated as a graph $G = (V, E)$, where $V$ denotes the agent set, $|V| = N$, and two agents can communicate with each other if there is an edge between them existing in $E$. A mixing matrix $W \in \mathbb{R}^{N \times N}$ is employed to encode the communication structure induced by the graph $G$.

**Definition 2.1** (Mixing Matrix). Let $G = (V, E)$ be an undirected connected graph with $|V| = N$ nodes. A matrix $W \in \mathbb{R}^{N \times N}$ is said to be a mixing matrix associated with the graph $G$ if it satisfies the following conditions: (i) $W$ is symmetric, i.e., $W = W^T$; (ii) $W$ is doubly stochastic, i.e.,

$W\mathbf{1} = \mathbf{1}$, where $\mathbf{1}$ denotes the all-ones column vector of an appropriate dimension; (iii) $0 \leq W_{ij} \leq 1$ for all $i, j \in [N]$; and (iv) $W_{ij} = 0$ if $(i, j) \notin E \cup \{(i, i)\}$.

**Definition 2.2** (Convexity). A function $f : \mathbb{R}^d \to \mathbb{R}$ is strongly convex with constant $\mu > 0$ if there exists a constant $\mu > 0$ such that for all $x, y \in \mathbb{R}^d$, $f(y) \geq f(x) + \nabla f(x)^\top (y - x) + \frac{\mu}{2} \|y - x\|^2$ holds. $f$ is convex if and only if $f(y) \geq f(x) + \nabla f(x)^\top (y - x)$ holds for all $x, y \in \mathbb{R}^d$.

**Definition 2.3** (Lipschitz Smoothness). A function $f : \mathbb{R}^d \to \mathbb{R}$ is Lipschitz smooth with constant $L > 0$ if for all $x, y \in \mathbb{R}^d$, there exists a constant $L$ such that the following inequality holds: $\|\nabla f(x) - \nabla f(y)\| \leq L \|x - y\|$.

## 2.2 PEP for Distributed Optimization with Local Step Sizes

For the convenience of presentation, we define the augmented state as $\boldsymbol{x}^k = [x_1^k, \ x_2^k, \ \ldots, \ x_N^k] \in \mathbb{R}^{d \times N}$, where the superscript $k$ belongs to the index set $I_K = \{0, \ldots, K\}$. We also define the augmented local optimum and global optimum as $\boldsymbol{x}^\star = [x_1^\star, \ x_2^\star, \ \ldots, \ x_N^\star]$ and $\boldsymbol{x}^* = [x^*, \ x^*, \ \ldots, \ x^*]$, respectively (where $x^* \in \mathbb{R}^d$ denotes a solution to (1)). In order to streamline the notation, we augment the index set $I_K$ by including the two optimal states, resulting in the extended index sets $I_K^\star = \{0, \ldots, K, \star\}$ and $I_K^{\star,*} = \{0, \ldots, K, \star, *\}$.

Analogously, we define the augmented gradients and function values as

$$\boldsymbol{g}^k = [g_1^k, \ g_2^k, \ \ldots, \ g_N^k] \in \mathbb{R}^{d \times N}, \quad \boldsymbol{f}^k = [f_1^k, \ f_2^k, \ \ldots, \ f_N^k] \in \mathbb{R}^N, \quad \text{for } k \in I_K^{\star,*}.$$

We use $\mathcal{F}_{\mu,L}$ to denote the class of functions in which each $f \in \mathcal{F}_{\mu,L}$ is $\mu$-strongly convex and $L$-smooth. Further denoting $\mathbb{S}_+$ as the set of all symmetric positive semidefinite matrices, we have

$$P = [\boldsymbol{x}^0, \ \boldsymbol{g}^0, \ \boldsymbol{g}^1, \ \ldots, \ \boldsymbol{g}^K, \ \boldsymbol{g}^\star, \ \boldsymbol{g}^*, \ \boldsymbol{x}^\star, \ \boldsymbol{x}^*] \in \mathbb{R}^{d \times [(K+6)N]},$$

$$G = P^\top P \in \mathbb{S}_+^{((K+6)N) \times ((K+6)N)}, \ F = [\boldsymbol{f}^0, \ \boldsymbol{f}^1, \ \ldots, \ \boldsymbol{f}^K, \ \boldsymbol{f}^\star, \ \boldsymbol{f}^*] \in \mathbb{R}^{1 \times (K+3)N},$$

which allows us to formulate the PEP as:

$$\max_{\substack{\boldsymbol{x}^0, \boldsymbol{g}^0, \boldsymbol{g}^1, \ldots, \boldsymbol{g}^K, \boldsymbol{g}^\star, \boldsymbol{g}^*, \boldsymbol{x}^\star, \boldsymbol{x}^* \\ \boldsymbol{f}^0, \boldsymbol{f}^1, \ldots, \boldsymbol{f}^K, \boldsymbol{f}^\star, \boldsymbol{f}^*}} \quad \frac{1}{N} \sum_{i=1}^N \|x_i^K - x^*\|^2 \tag{2}$$

s.t. $\{x_i^k, \ g_i^k, \ f_i^k\}, \ k \in I_K^\star$ and $\{x^*, \ g_i^*, \ f_i^*\}$ are interpolated for each local function class $\mathcal{F}_{\mu_i, L_i}$, \tag{3}

$\{x_i^k\}, \ i \in [N], \ k \in I_K$ are generated recursively by method $\mathcal{M}$, \tag{4}

$\sum_{i=1}^N g_i^* = \sum_{i=1}^N \nabla f_i(x^*) = 0$, such that $x^*$ is the global optimum, \tag{5}

$g_i^\star = \nabla f_i(x_i^\star) = 0, \ \forall i \in [N]$, such that $x_i^\star$ is the local optimum for $f_i$, \tag{6}

$\|x_i^0 - x^*\|^2 \leq R_0^2, \ \forall i \in [N], \quad \|x_i^\star - x^*\|^2 \leq R_*^2, \ \forall i \in [N]$. \tag{7}

Constraint (3) guarantees that the points generated by the algorithm can be interpolated by a function in the corresponding functional class. Constraint (4) require the local states of the algorithm to be linear combinations of the previously defined PEP variables. Finally, (5) ensures that $x^*$ is an optimal solution to problem (1), while (6) guarantees that each $x_i^\star$ is a local optimum of $f_i$ for all $i \in [N]$.

The key challenge in making the aforementioned PEP a solvable SDP is that the constraints in (3) are infinite-dimensional. Fortunately, this can be resolved with the following lemma:

**Lemma 2.1.** (Interpolation for function class $\mathcal{F}_{\mu,L}$(Taylor et al., 2016)) Let $I$ be any interpolation index set, and let $\{x^k, g^k, f^k\}_{k \in I} \subset \mathbb{R}^d \times \mathbb{R}^d \times \mathbb{R}$. There exists $f \in \mathcal{F}_{\mu,L}$ satisfying $f(x^k) = f^k$ and $g^k \in \partial f(x^k)$ for all $k \in I$ if and only if the following inequality holds:

$$f_i(x_i) - f_j(x_j) - \langle g_j, x_i - x_j \rangle \geq$$
$$\frac{1}{2}\left(1 - \frac{\mu}{L}\right)\left[\frac{1}{L}\|g_i - g_j\|^2 + \mu\|x_i - x_j\|^2 - \frac{2\mu}{L}\langle g_j - g_i, x_j - x_i \rangle\right]. \tag{8}$$

In this way, the constraints in (3) can be reformulated into equivalent, finite-dimensional constraints, which makes the problem tractable and solvable using standard SDP solvers. Overall, our formulation differs from that of Colla and Hendrickx (2023) in three key aspects:

1. Our formulation results in a matrix $G$ whose dimension is one-third of that in Colla and Hendrickx (2023), allowing for more efficient computation and greater scalability with respect to $N$ and $K$. This reduction is crucial for our analysis in the multi-agent setting;

2. We use $\frac{1}{N} \sum_{i=1}^{N} ||x_i^K - x^*||^2$ as our measure for optimization performance, which captures both the consensus error and the optimality gap, whereas the measure $f(\bar{x}) - f^*$ used in Colla and Hendrickx (2023) only captures the optimality gap;

3. Instead of enforcing the constraint $\frac{1}{N} \sum_{i=1}^{N} ||s_i^0 - \frac{1}{N} \sum_{j=1}^{N} \nabla f_j(x_j^0)||^2 \leq E^2$ as done in Colla and Hendrickx (2023), where $s_i^0$ is the initial local guess on the global gradient and $E$ is predefined input, we follow Arjevani and Shamir (2015); Tian et al. (2022) and impose the constraint (7) on the distance between the local and global optima.

## 3 ALGORITHM DESIGN AND THEORETICAL RESULT

### 3.1 PEP ANALYSIS

**Theorem 3.1.** Consider the class $\mathcal{F}_{\mu_i, L_i}$ of $L_i$-smooth $\mu_i$-strongly convex functions, the performance criterion $\frac{1}{N} \sum_{i=1}^{N} ||x_i^K - x^*||^2$, and a fixed-step size distributed optimization method $\mathcal{M}$ that computes $K$ iterates according to mixing matrix $W \in \mathbb{R}^{N \times N}$. If $(K + 6)N \leq d$ holds, the **exact** worst-case optimization error after $K$ iterations of method $\mathcal{M}$ is equal to the optimal value of the semidefinite programming problem characterized by (2)-(7).

*Informal proof.* (see detailed version in the appendix) We can reformulate the problem characterized by (2)-(7) into an equivalent SDP (semidefinite programming). Constraints (3) are the necessary and sufficient conditions for $f_i \in \mathcal{F}_{\mu_i, L_i}$ $i \in [N]$ by lemma (2.1). Therefore, the optimal value of the SDP is the **exact** worst-case optimization error. $\qquad\square$

To efficiently solve the resulting SDPs arising from the PEP framework, we utilize the JuMP modeling language in Julia. JuMP (Lubin et al., 2023) provides a flexible and high-level interface for formulating convex optimization problems, including SDPs, and allows for seamless integration with state-of-the-art solvers. For this work, we employ the MOSEK solver (ApS, 2019), which is well-suited for large-scale and structured convex optimization problems, particularly semidefinite and conic programs. This combination enables us to model and solve the PEP instances accurately and efficiently. All experiments are conducted on a computing platform equipped with dual AMD EPYC 9654 processors (each with 96 cores) and 256 GB of RAM. Under this hardware configuration, the maximum feasible dimension of the variable matrix $G$ is approximately 200. Now we are ready to utilize the SDP-PEP framework to gain insight into the design of distributed optimization algorithms[1].

### 3.2 PEP RESULTS AND ALGORITHM DESIGN

We consider the EXTRA algorithm (Shi et al., 2015) and the NIDS algorithm (Li et al., 2019b), which allow agents to use larger step sizes than the gradient-tracking based distributed optimization algorithms (see, e.g., Xu et al. (2015); Nedic et al. (2017); Pu et al. (2020)). It is worth noting that we do not consider the distributed gradient descent algorithm in Nedic and Ozdaglar (2009b) since it cannot ensure both fast and accurate convergence (it can only provide accurate convergence in distributed optimization under a diminishing step size, which, however, slows down convergence). Under smooth and convex objective functions, the NIDS algorithm reduces to $x^{k+1} = \tilde{W} \left( 2x^k - x^{k-1} - \Lambda g(x^k) + \Lambda g(x^{k-1}) \right)$, where $\tilde{W} = I - c\Lambda(I - W)$, $0 < c \leq \frac{1}{(1-\lambda_N(W)) \max_i \alpha_i}$, $\lambda_N(W)$ denotes the $N$-th largest eigenvalue of $W$, and $\Lambda = \text{diag}(\alpha_1, ..., \alpha_N)$. The dynamics of EXTRA are given by $x^{k+1} = \frac{I+W}{2} \left( 2x^k - x^{k-1} - \Lambda g(x^k) + \Lambda g(x^{k-1}) \right)$ [2]. The performance of the two algorithms is summarized by the blue line and green line, respectively, in Figure 1. It is worth noting that for NIDS, we only present the results with a global step size $\alpha = \frac{1}{L}$ since NIDS with local step sizes $\alpha_i = \frac{1}{L_i}$ converges slower. It is also worth noting that EXTRA with a global step size

---

[1] Code is available at https://anonymous.4open.science/r/localstepsize-6B15/README.md

[2] This is slightly different with the original EXTRA (Shi et al., 2015), where the update is $x^{k+1} = \frac{I+W}{2} \left( 2x^k - x^{k-1} \right) - \alpha g(x^k) + \alpha g(x^{k-1})$.

$\alpha = \frac{1}{L}$ has the same dynamics as NIDS with the same global step size. In the results, we consider the same set of function class $F_{\mu, L_i}, \forall i \in [N]$.

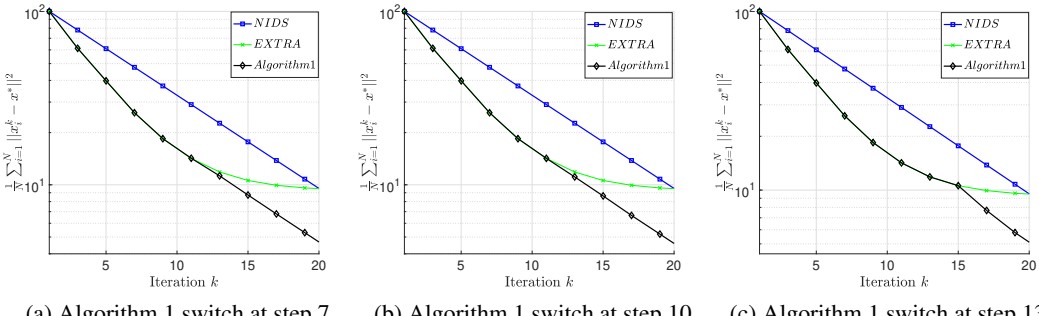

(a) Algorithm 1 switch at step 7     (b) Algorithm 1 switch at step 10     (c) Algorithm 1 switch at step 13

Figure 1: Performance comparison of EXTRA, NIDS, and Algorithm 1 using PEP. The performance metric is the optimization error $\frac{1}{N} \sum_{i=1}^{N} \|x_i^K - x^*\|^2$. The switching time instant for Algorithm 1 was set to 7, 10, and 13, respectively, in subplots (a), (b), and (c). The blue curve represents NIDS with a global step size $\frac{1}{\bar{L}}$ with $\bar{L} = \frac{1}{N} \sum_{i=1}^{N} L_i$, which has the same update dynamics as Algorithm 1 under the same global step size. The black curve represents the performance of Algorithm 1 with local step sizes. The green curve represents the performance of the EXTRA algorithm with local step sizes, i.e., agent $i$ using a step size $\frac{1}{L_i}$. In the result, we consider two agents with heterogeneous local Lipschitz constants $L_1 = \frac{1}{3}$ and $L_2 = 3$.

---

**Algorithm 1** (From the perspective of agent $i$)

---

1: **Initialization** $x_i^0$, step size $\alpha_i, c, x_i^1 = x_i^0 - \alpha_i \nabla f_i(x_i^0), \tilde{W}_{ii} = \frac{W_{ii}+1}{2}$ and $\tilde{W}_{ii} = W_{ij}$ for $j \neq i$, where $\tilde{W}_{ij}$ and $W_{ij}$ denotes the $ij$-th entry of $\tilde{W}$ and $W$, respectively.
2: **for** $k = 1, ..., K$ **do**
3:     **if** $\mathcal{S}$ is True **then**
4:        Re-initialize, and set $\tilde{W}_{ii} = 1 - c\alpha_i + c\alpha_i W_{ii}$ and $\tilde{W}_{ij} = c\alpha_i W_{ij}$ for $j \neq i$.
5:     **end if**
6:     Compute local gradient $g_i^k = \nabla f_i(x_i^k)$ and $x_i^{k+\frac{1}{2}} = 2x_i^k - x_i^{k-1} - \alpha_i g_i^k + \alpha_i g_i^{k-1}$;
7:     Share $x_i^{k+\frac{1}{2}}$ with neighbors and update $x_i^{k+1} = \sum_{j=1}^{N} \tilde{W}_{ij} x_i^{k+\frac{1}{2}}$.
8: **end for**

---

The results in Figure 1 demonstrate that NIDS can ensure accurate convergence; however, its convergence speed is relatively slow. Conversely, EXTRA can provide faster convergence but is subject to a final optimization error. To ensure both fast and accurate convergence, we propose Algorithm 1 to combine EXTRA and NIDS. The idea is to use EXTRA and local step sizes in the early state of optimization; when the convergence speed plateaus (can be reflected by, e.g., diminishing variation in gradient values), we switch to NIDS with local step sizes, which can ensure accurate convergence. The performance of our Algorithm 1 is presented by the black curve in Figure 1. It is clear that it converges faster than both EXTRA and NIDS while ensuring accurate convergence. It is worth noting that Algorithm 1 has theoretically guaranteed convergence:

**Theorem 3.2.** (Convergence of Algorithm 1) Consider the distributed optimization problem (1) where each $f_i$ is $\mu_i$ strongly convex and $L_i$ smooth, and the mixing matrix $W$ satisfies the conditions specified in Definition 2.1. Suppose the step sizes satisfy $\alpha_i \leq \frac{2}{L_i}$ for all $i \in [N]$, and $c$ is chosen such that $I - c\Lambda^{1/2}(I - W)\Lambda^{1/2} \succeq 0,$. Let $\{x_i^K\}_{i \in [N]}$ be the iterates generated by Algorithm 1. Then for all $K \geq 1$, we have

$$\frac{1}{N} \sum_{i=1}^{N} \frac{1}{\alpha_i} \|x_i^K - x^*\|^2 \leq \rho^K C(\mathcal{S}, \{x_i^0\}_{i \in [N]}, \{\alpha_i\}_{i \in [N]}),$$

where

$$\rho = \max\left(1 - \left(2 - \max_{i \in [N]}(\alpha_i L_i)\right) \min_{i \in [N]}(\mu_i \alpha_i), \ 1 - c \lambda_{\max}\left(\Lambda^{-1/2}(I - W)^{\dagger}\Lambda^{-1/2}\right)\right) \in (0, 1),$$

and $C(\mathcal{S}, \{x_i^0\}_{i\in[N]}, \{\alpha_i\}_{i\in[N]})$ is a constant depending on the switching condition $\mathcal{S}$, the initialization $\{x_i^0\}_{i\in[N]}$, and the step sizes $\{\alpha_i\}_{i\in[N]}$.

*Proof.* See Appendix 6.3.4. □

It is worth noting that to test the sensitivity of Algorithm 1 to the switching time instant, we plot the performance of Algorithm 1 under different switching time instants. The results show that Algorithm 1 consistently outperforms EXTRA and NIDS, even when the switching time is not carefully tuned.

**Remark 3.1.** In Algorithm 1, $\mathcal{S}$ denotes the switching condition under which the update rule transitions from EXTRA to NIDS. The problem of determining the switch time is essentially **equivalent to** the universal problem of determining stopping times in all decentralized iterative algorithms, which has been well-studied in distributed optimization and control (Manitara et al., 2022; Rikos et al., 2025). Consequently, existing approaches for decentralized stopping can be **adapted to our setting with little modification**. For example, the switching time can be predetermined (e.g., after a fixed number of iterations) or determined adaptively through detection mechanisms that monitor convergence. For example, each agent may track its local convergence behavior by monitoring the norm of its gradient or loss. If the change in this value remains below a predefined threshold for several consecutive iterations, the agent concludes that progress has stalled, and broadcasts a 1-bit switch signal to its neighbors. Using the standard flooding technique (Kuruzov et al., 2024), this signal is rapidly propagated to all agents within at most $N-1$ communication rounds, even under unknown or dynamically changing network topologies, while incurring negligible communication overhead. Once every agent receives the signal, the switch is executed globally. If multiple agents initiate flooding concurrently, the result remains the same: all agents switch exactly once. Thus, the switching process occurs only once during the entire runtime of the algorithm and can be implemented efficiently, as it requires only the exchange of 1-bit messages. In fact, as shown in Figure 1, even an imperfect switching mechanism does not prevent Algorithm 1 from achieving faster convergence than both NIDS and EXTRA.

**Remark 3.2.** When using a global step size $\frac{1}{\bar{L}}$, where $\bar{L} = \frac{1}{N}\sum_{i=1}^{N} L_i$, Algorithm 1 degenerates to EXTRA or NIDS: $x^{k+1} = \tilde{W}\left(2x^k - x^{k-1} - \frac{1}{\bar{L}}\nabla g(x^k) + \frac{1}{\bar{L}}\nabla g(x^{k-1})\right)$. Hence, comparing with NIDS corresponds to comparing with Algorithm 1 with a global step size.

### 3.3 INFLUENCE OF FUNCTION HETEROGENEITY ON ALGORITHM 1'S SPEED GAIN OVER EXTRA AND NIDS

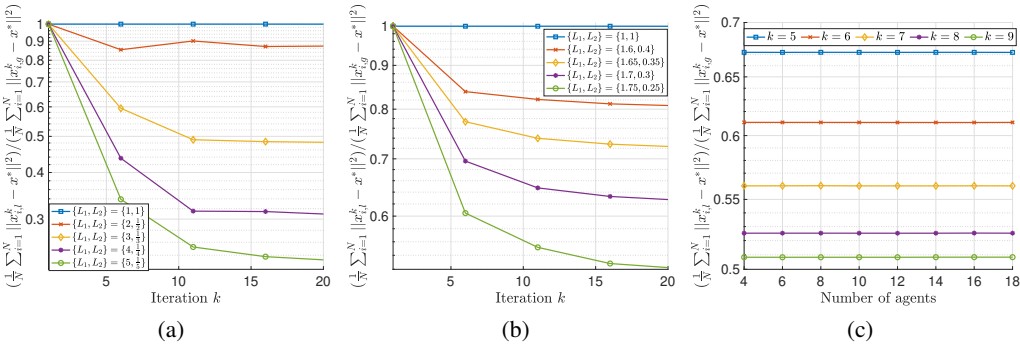

(a)    (b)    (c)

Figure 2: Convergence speed gain of Algorithm 1 with local step sizes over Algorithm 1 with global step sizes (which reduces to EXTRA and NIDS). The y-axis shows the ratio of optimization errors obtained by PEP under the local step size strategy of $\alpha_i = \frac{1}{L_i}$ for agent $i$ versus the global step size strategy of $\alpha = \frac{1}{\bar{L}}$ with $\bar{L} = \frac{1}{N}\sum_{i=1}^{N} L_i$. A ratio less than 1 indicates superior convergence performance of the local step size strategy to its global step size counterpart. (a) and (b) consider the two-agent setting under heterogeneous Lipschitz constants satisfying $L_1 * L_2 = 1$ in (a) and $L_1 + L_2 = 2$ in (b). (c) considers the more than two-agent setting with half of the agents having a Lipschitz constant of $\frac{1}{3}$, whereas the other half have a Lipschitz constant of 3. The interaction topology is all-to-all, and the strongly convex parameter is always $\mu = 0.1$.

In this section, we characterize the speed gain of Algorithm 1 from local step sizes over EXTRA and NIDS under a global step size (in this case, EXTRA and NIDS have the same dynamics). To quantify the speed gain, we use the ratio of **exact** optimization error between Algorithm 1 with local step sizes and the version where a global step size is used. The following theorem establishes the advantage of Algorithm 1 in convergence speed, leveraging local step sizes, over EXTRA and NIDS.

**Theorem 3.3.** (Convergence Acceleration of Local Step sizes) Consider the distributed optimization problem defined in (1), where the global function $f$ belongs to the class $\mathcal{F}_{\mu,L}$, characterized by the smoothness parameter $L$ and strong convexity parameter $\mu$. Let $\mathcal{A}_l$ denote Algorithm 1 with local step sizes $\frac{1}{L_i}$, and $\mathcal{A}_g$ denote Algorithm 1 with a global step size $\frac{1}{L}$ (which reduces to EXTRA and NIDS). For the performance metric

$$P_{\mathcal{A}} := \max_{f \in \mathcal{F}_{\mu,L}} \left\{ \frac{1}{N} \sum_{i=1}^{N} \|x_{i,\mathcal{A}}^k - x^*\|^2 \right\},$$

which measures the worst-case optimization error at iteration $k$ over all $f \in \mathcal{F}_{\mu,L}$. Consider the ratio $P_{\mathcal{A}_l}/P_{\mathcal{A}_g}$ as the relative acceleration of the local step sizes method compared with the global step size method (with values below 1 indicating acceleration and smaller ratios indicating stronger acceleration). Figure 2 gives the exact ratio values under different function classes, and the ratio values being strictly less than 1 for all tested function classes $\mathcal{F}_{\mu,L}$ in the heterogeneous setting confirm that Algorithm 1 with local step sizes achieves a strictly faster convergence rate than its global step size counterpart under all considered function classes.

*Proof.* By Theorem 3.1, we have $P_{\mathcal{A}} < \infty$, which implies that the worst-case optimization error of Algorithm 1 over the function class $\mathcal{F}_{\mu,L}$ is always finite. Moreover, the set $\arg\max_{f \in \mathcal{F}_{\mu,L}} \left\{ \frac{1}{N} \sum_{i=1}^{N} \|x_{i,\mathcal{A}}^k - x^*\|^2 \right\}$ is guaranteed to be nonempty. This means that there exists at least one function instance $f_{\mathcal{A}}^*$ for which Algorithm 1, when run with step-size scheme $\mathcal{A}$, actually attains its worst-case performance at iteration $k$. For any such maximizer $f_{\mathcal{A}}^* \in \arg\max_{f \in \mathcal{F}_{\mu,L}} \left\{ \frac{1}{N} \sum_{i=1}^{N} \|x_{i,\mathcal{A}}^k - x^*\|^2 \right\}$, the value $P_{\mathcal{A}}$ returned by the PEP is therefore an *exact*, not merely upper-bounding, characterization of the true worst-case error over $\mathcal{F}_{\mu,L}$. In other words, the PEP does not produce a loose bound: it identifies a specific problem instance on which Algorithm 1 performs exactly as poorly as the bound indicates.

With this interpretation, the ratio $\frac{P_{\mathcal{A}_L}}{P_{\mathcal{A}_g}}$ directly compares the exact worst-case optimization errors of Algorithm 1 when using local step sizes versus using a global step size. A value $\frac{P_{\mathcal{A}_L}}{P_{\mathcal{A}_g}} < 1$ therefore means that—in the worst case—using local step sizes leads to a strictly smaller optimization error at iteration $k$. Hence, the ratio values in Figure 2 quantitatively capture the exact speed gains achieved by adopting local step sizes in Algorithm 1. □

It is clear that the smaller the ratio is, the bigger the speed gain is. Figure 2(a) shows the speed gain of Algorithm 1 under two agents with different local Lipschitz constants $L_1$ and $L_2$ satisfying $L_1 * L_2 = 1$ (it is worth noting that the result is invariant when the two agents swap $L_1$ and $L_2$). The interaction topology was set to $W = \frac{\mathbf{1}\mathbf{1}^T}{N}$, and the switching time instant was set to 10. It is clear that higher heterogeneity in local Lipschitz constants leads to more speed gains. Figure 2(b) shows that the same conclusion can be drawn when $L_1$ and $L_2$ have a fixed sum value. Figure 2(c) shows the influence of the network size on the ratio. For the convenience of comparison, we let half of the agents have a Lipchitz constant equal to $L_1$ and the other half of the agents have a Lipchitz constant equal to $L_2$. It is clear that the ratio, or the speed gain of Algorithm 1, is not sensitive to the size of the network, which is consistent with the observations in Colla and Hendrickx (2024). Additional results in the Appendix confirm that the same conclusion can be drawn under other mixing matrices.

## 4 EXPERIMENTS

We compare Algorithm 1 under local step sizes versus under a global step size (which reduces to EXTRA and NIDS that have the same dynamics under a global step size) using two regression tasks[3].

---

[3]Code is available at https://anonymous.4open.science/r/localstepsize-6B15/README.md

### 4.1 RIDGE REGRESSION

Ridge regression (van Wieringen, 2015) is a regularized extension of linear regression that mitigates multicollinearity and overfitting by incorporating an $\ell_2$ penalty on the regression coefficients. Given an input matrix $X_i \in \mathbb{R}^{m \times n}$ and a label vector $y_i \in \mathbb{R}^m$, ridge regression solves the following optimization problem: $\min_{w \in \mathbb{R}^d} \sum_{i=1}^{N} f_i(\omega)$, where $f_i(\omega) = \|X_i w - y_i\|_2^2 + \frac{\mu}{2}\|w\|_2^2$ and $\mu > 0$ is a regularization parameter. To generate random samples with different Lipschitz constants, we exploit the Hessian matrix of $f_i$, which is given by $\nabla^2 f_i(\omega) = 2X_i^T X_i + \mu I$. The Hessian matrix indicates that the Lipschitz constant is determined by $\lambda_{\max}(2X_i^T X_i + \mu I)$ and the strongly convex parameter is determined by $\lambda_{\min}(2X_i^T X_i + \mu I)$. Using this property, we can generate $X_i$ and corresponding $y_i$ with given Lipschitz and strongly convex parameters (Kovalev et al., 2022). In our experiments, we consider $m = 100, n = 50$ and three different data distribution scenarios. In the first scenario, we consider a network of two agents with local Lipschitz constants given by 1 and 2, respectively. In the second scenario, we consider a network of six agents divided into two groups with the Lipschitz constant of each group given by 1 and 2, respectively. And in the third scenario, we consider 6 agents with heterogeneous Lipschitz constants given by 1, 1.4, 1.2, 2.2, 2.7, and 2.3, respectively. In all scenarios, the strongly convex parameter was set to 0.01 without loss of generality. The results for the three scenarios are summarized in Figure 3(a), Figure 3(b), and Figure 3(c), respectively. It can be seen that local step sizes can always converge faster than the case with a global step size. Additionally, we conducted comparisons with DGD (Nedic and Ozdaglar, 2009a) and SONATA (Sun et al., 2022), for which the step sizes are tuned to achieve their best observed performance. The results are also depicted in Figure 3.

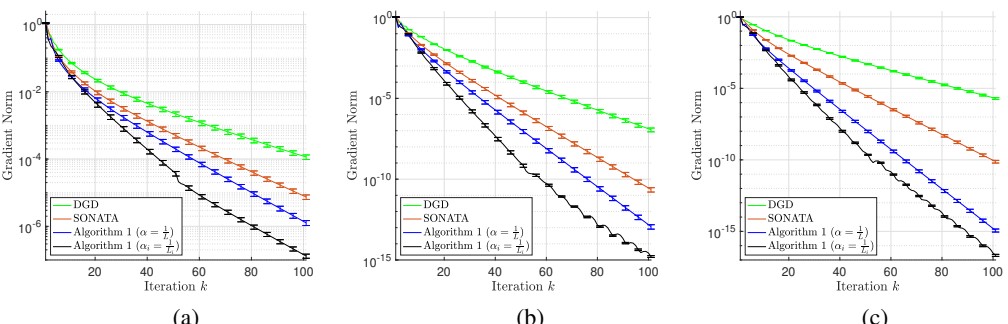

(a)  (b)  (c)

Figure 3: Comparison of Algorithm 1 with local step sizes ($\alpha_i = \frac{1}{L_i}$), its global step-size counterpart ($\alpha = \frac{1}{L}$, reducing to EXTRA and NIDS), and the methods DGD (Nedic and Ozdaglar, 2009a) and SONATA (Sun et al., 2022). In (a), we consider a two-agent setting with heterogeneous Lipschitz constants $\{1, 2\}$. In (b), we consider six agents, with half of them having a local Lipschitz constant of 1 and the other half having a local Lipschitz constant of 2. In (c), the six agents have distinct Lipschitz constants 1, 1.4, 1.2, 2.2, 2.7 and 2.3, respectively. The interaction pattern is an all-to-all graph. All the experiments are run independently 5 times with random initialization on a sphere. The results show that in all cases, using local step sizes (agent $i$ using $\frac{1}{L_i}$) leads to a substantial increase in convergence speed compared with the case where a global step size $\frac{1}{L}$ is used for all agents.

### 4.2 LOGISTIC REGRESSION ON REAL DATA

We also evaluate Algorithm 1 on the benchmark w8a dataset, which is part of the LIBSVM (Chang and Lin, 2011) collection. The local functions are given by

$$f_i(w) = \frac{1}{n}\sum_{j=1}^{n} \log\left(1 + \exp(-y_j x_j^T w)\right) + \frac{\mu}{2}\|w\|_2^2,$$

where $x_j \in \mathbb{R}^d$ and $y_j \in \{-1, 1\}$ are data samples. We use a simple two-agent setting to compare the performance of Algorithm 1 under local step sizes and Algorithm 1 under a global step size (which reduces to EXTRA and NIDS that have the same dynamics under a global step size). We consider three different data splitting approaches between the two agents which result in different Lipschitz properties of the two local objective functions (our results in Section 3 show that the speed gain of Algorithm 1 is bigger when the step size heterogeneity is bigger, so we split the data in non-uniform

manners to ensure large heterogeneity between the two agents' local Lipschitz constants). In the first splitting approach, we divide the data according to the label value $y_j$. In the second splitting approach, we order the magnitude of $\|x_j\|^2$ and assign the smaller half of the samples to agent 1 and the other half to agent 2. In the third splitting approach, we order the maximal eigenvalues of $x_j x_j^T$ and assign the smaller half samples to agent 1 and the rest to agent 2. In all splitting cases, we expect the two agents to have different Lipschitz constants. The results are summarized in Figure 4(a), Figure 4(b), and Figure 4(c), respectively. It can be seen that in all cases, Algorithm 1 with local step sizes (agent $i$ using $\alpha_i = \frac{1}{L_i}$) has faster convergence than the case with a global step size (all agents use the same step size $\alpha = \frac{1}{\bar{L}}$ with $\bar{L} = \frac{1}{N}\sum_{i=1}^{N} L_i$). Additionally, we also conducted comparisons with DGD (Nedic and Ozdaglar, 2009a) and SONATA (Sun et al., 2022), for which the step sizes are tuned to achieve their best observed performance. The results are depicted in Figure 4.

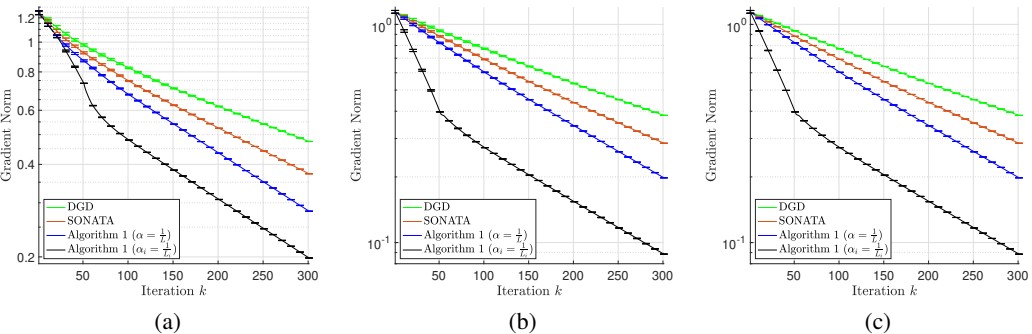

Figure 4: Comparison of Algorithm 1 with local step sizes ($\alpha_i = \frac{1}{L_i}$), its global step-size counterpart ($\alpha = \frac{1}{\bar{L}}$, reducing to EXTRA and NIDS), and the methods DGD (Nedic and Ozdaglar, 2009a) and SONATA (Sun et al., 2022) using the w8a dataset. The three subplots correspond to the results obtained using three different subset selection criteria. All experiments are conducted independently 3 times, each with a random initialization on a sphere. Subplot (a) represents the case where the data are split into two agents according to the labels $y_j$, a strategy that is straightforward and inexpensive to implement. Subplot (b) corresponds to the case where the data are split into two agents according to the squared norms $\|x_j\|^2$, which is comparatively easier to compute. Subplot (c) illustrates the scenario where the subsets of the data are split between two agents according to the eigenvalues of $x_j x_j^T$ for each data sample $j$, a procedure that is relatively computationally intensive. The results show that in all cases, using local step sizes (agent $i$ using $\frac{1}{L_i}$) leads to a substantial increase in convergence speed compared with the case where a global step size $\frac{1}{\bar{L}}$ is used by all agents.

## 5    CONCLUSION

In this work, we present the first rigorous and systematic evidence that local step sizes can accelerate distributed optimization compared with the case where a universal global step size is used for all agents. The results are obtained by generalizing the semidefinite programming based performance characterizing framework by incorporating practical constraints and improving its computational efficiency. They also inspired us to develop a new algorithm that can exploit local step sizes to yield faster convergence than the state-of-the-art distributed optimization algorithms EXTRA and NIDS. Besides SDP based rigorous performance characterization, we also confirm the theoretical findings using regression results on benchmark datasets. These results underscore the importance of leveraging local data structures in designing distributed algorithms and pave the way for further exploring data-aware strategies in distributed learning and optimization.

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

## 6 APPENDIX

### 6.1 PROOF OF THE THEOREM 3.1

In this section, we prove (3.1) by formulating the Performance Estimation Problem (PEP) characterized by (2)-(7) as an equivalent SDP. We use the standard notation $e_i \in \mathbb{R}^d$ to denote the unit $d$-dimensional vector with a nonzero entry "1" in the $i$-th component. The following vectors are employed to select specific columns in $P$ and $F$:

$$\mathbf{f}_i^k = e_{kN+i} \in \mathbb{R}^{(K+3)N}, \mathbf{f}_i^\star = e_{(K+1)N+i} \in \mathbb{R}^{(K+3)N}, \mathbf{f}_i^* = e_{(K+2)N+i} \in \mathbb{R}^{(K+3)N},$$

$$\mathbf{g}_i^k = e_{(k+1)N+i} \in \mathbb{R}^{(K+6)N}, \mathbf{g}_i^\star = e_{(K+2)N+i} \in \mathbb{R}^{(K+6)N}, \mathbf{g}_i^* = e_{(K+3)N+i} \in \mathbb{R}^{(K+6)N},$$

$$\mathbf{x}_i^0 = e_i \in \mathbb{R}^{(K+6)N}, \mathbf{x}_i^\star = e_{(K+4)N+i} \in \mathbb{R}^{(K+6)N}, \mathbf{x}_i^* = e_{(K+5)N+i} \in \mathbb{R}^{(K+6)N}, \forall i \in [N], k \in I_K.$$

The stacked version of these vectors is analogous to the formulation in section (2.2). Now we are in a position to give our formulation of the PEP for a decentralized algorithm $\mathcal{M}$ as follows

$$\max_{F,G} \quad \langle G, (\sum_{i=1}^{N}(\mathbf{x}_i^K - \mathbf{x}_i^*))(\sum_{i=1}^{N}(\mathbf{x}_i^K - \mathbf{x}_i^*)^T)\rangle \tag{9}$$

$$\text{subject to} \quad \langle G, A(\{\mathbf{x}_i^k, \mathbf{g}_i^k, \mathbf{f}_i^k\}_{k\in\{p,q\}}, \mu_i, L_i)\rangle \leq F * (\mathbf{f}_i^p - \mathbf{f}_i^q), \ \forall i \in [N], \ p, q \in I_K^{\star,*}, \tag{10}$$

$$\{\mathbf{x}_i^k\}_{i\in[N], k\in I_K} \text{ are generated recursively by algorithm } \mathcal{M}, \tag{11}$$

$$\langle G, (\mathbf{x}_1^* - \mathbf{x}_i^*)(\mathbf{x}_1^* - \mathbf{x}_i^*)^T\rangle = 0, \ \forall i \in [N], \tag{12}$$

$$\langle G, (\sum_{i=1}^{N}\mathbf{g}_i^*)(\sum_{i=1}^{N}\mathbf{g}_i^*)^T\rangle = 0, \quad \langle G, \mathbf{g}_i^\star \mathbf{g}_i^{\star T}\rangle = 0, \ \forall i \in [N], \tag{13}$$

$$\langle G, (\mathbf{x}_i^0 - \mathbf{x}_i^*)(\mathbf{x}_i^0 - \mathbf{x}_i^*)^T\rangle \leq R_0^2, \ \forall i \in [N], \tag{14}$$

$$\langle G, (\mathbf{x}_i^\star - \mathbf{x}_i^*)(\mathbf{x}_i^\star - \mathbf{x}_i^*)^T\rangle \leq R_*^2, \ \forall i \in [N], \tag{15}$$

$$G \succeq 0, \tag{16}$$

$$\text{rank}(G) \leq d, \tag{17}$$

where

$$A(\{\mathbf{x}_i^k, \mathbf{g}_i^k, \mathbf{f}_i^k\}_{k\in\{p,q\}}, \mu_i, L_i)$$

$$= \frac{1}{2}\left[(\mathbf{x}_i^q - \mathbf{x}_i^p)\mathbf{g}_i^{qT} + \mathbf{g}_i^q(\mathbf{x}_i^q - \mathbf{x}_i^p)^T\right] - \frac{\mu_i(L_i - \mu_i)}{2L_i^2}\left[(\mathbf{x}_i^q - \mathbf{x}_i^p)(\mathbf{g}_i^q - \mathbf{g}_i^p)^T + (\mathbf{g}_i^q - \mathbf{g}_i^p)(\mathbf{x}_i^q - \mathbf{x}_i^p)^T\right]$$

$$+ \frac{1}{2}\left(1 - \frac{\mu_i}{L_i}\right)\left[\frac{1}{L_i}(\mathbf{g}_i^p - \mathbf{g}_i^q)(\mathbf{g}_i^p - \mathbf{g}_i^q)^T + \mu_i(\mathbf{x}_i^p - \mathbf{x}_i^q)(\mathbf{x}_i^p - \mathbf{x}_i^q)^T\right],$$

and $\langle\cdot,\cdot\rangle$ denotes the standard matrix inner product defined as $\langle A, B\rangle = \text{trace}(AB^T)$. (9) represents the measure $\sum_{i=1}^{N}\|x_i^K - x^*\|^2$. Constraints (10) are the necessary and sufficient conditions for smooth, strongly convex interpolation; that is, the points indexed by $I$ can be interpolated by a function satisfying all constraints in (10) for all $i, j \in I$. So the infinite dimensional function constraint $f_i \in \mathcal{F}_{\mu_i, L_i}$ can now be expressed in terms of all the iterate and optimal points $\{x_i^k, g_i^k, f_i^k\}_{i\in[N], k\in I_K^\star}$ and $\{x^*, g_i^*, f_i^*\}_{i\in[N]}$ only, using the interpolation conditions given above. Constraints (10) corresponds to the interpolation constraints of Lemma (2.1), where we use the simple identity $\langle g_j, x_i - x_j\rangle = \frac{1}{2}\langle g_j, x_i - x_j\rangle + \frac{1}{2}\langle x_i - x_j, g_j\rangle$. Consequently, the resulting Gram matrix expression

$$\frac{1}{2}\mathbf{g}_j^\top P^\top P(\mathbf{x}_i - \mathbf{x}_j) + \frac{1}{2}(\mathbf{x}_i - \mathbf{x}_j)^\top P^\top P\mathbf{g}_j = \left\langle G, \frac{1}{2}\left((\mathbf{x}_i - \mathbf{x}_j)\mathbf{g}_j^\top + \mathbf{g}_j(\mathbf{x}_i - \mathbf{x}_j)^\top\right)\right\rangle$$

holds, where the matrix $\frac{1}{2}\left((\mathbf{x}_i - \mathbf{x}_j)\mathbf{g}_j^\top + \mathbf{g}_j(\mathbf{x}_i - \mathbf{x}_j)^\top\right)$ is symmetric. This technique enables us to reformulate Lemma (2.1) into the form of (10) for all points generated during the iteration process. Constraint (11) arises from the fact that, for each agent $i$, we have the following relation: $\mathbf{x}_i^K \in \text{Span}\left\{\{\mathbf{x}_i^k, \mathbf{g}_i^k, \mathbf{f}_i^k\}_{k\in I_K, i\in[N]}\right\}$, meaning that $\mathbf{x}_i^K$ is a linear combination of the elements $\{\mathbf{x}_i^k, \mathbf{g}_i^k, \mathbf{f}_i^k\}_{k\in I_K, i\in[N]}$ generated by the algorithm $\mathcal{M}$. Therefore, once the initial point is fixed, we

can recursively determine $\mathbf{x}_i^K$ for all $i \in [N]$. Constraints (13) ensure that $x^*$ is one of the global optimum and $x_i^\star$ is the local optimum of each $f_i$. For constraints (14) and (15), $R_*$ bounds the distance between local and global optima, i.e., $\|x_i^\star - x^*\| \le R_*$, $R_0$ bounds the distance between the initial point and the global optimum, i.e., $\|x_i^0 - x^*\| \le R_0$, and generally $R_* \le R_0$.

According to Taylor et al. (2016), the optimization problem characterized by (9)-(17), along with its solution, is independent of the dimension $d$. Therefore, the rank constraint (17) is redundant and can be omitted. After removing (17), the optimization problem characterized by (9)-(16) becomes an SDP.

## 6.2 ADDITIONAL EXPERIMENTS

**Ridge Regression** We provide additional distributed optimization experiments for 10-agent under several standard sparse graphs. The graph specifications and corresponding results are summarized in Figure 5. These results consistently demonstrate the advantage of using local step sizes compared with using a single global step size.

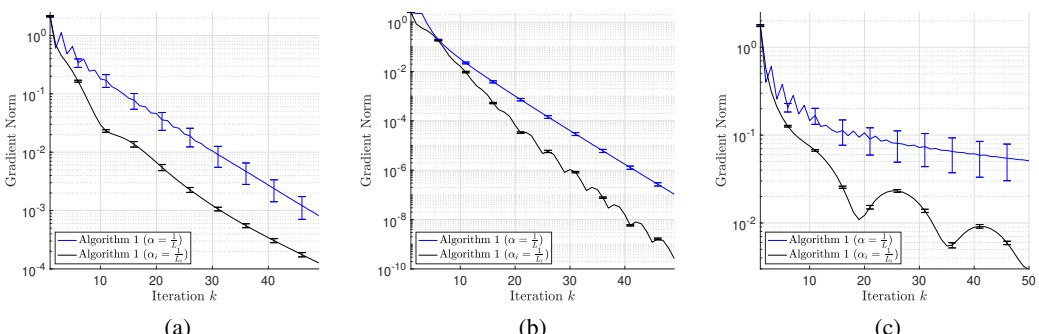

Figure 5: Comparison of Algorithm 1 with local step sizes and Algorithm 1 with a global step size (which reduces to EXTRA and NIDS that have the same dynamics under a global step size). The three subplots correspond to the results obtained using three different sparse graphs. All experiments are conducted independently 5 times, each with a random initialization on a sphere. Subplot (a) represents the case when the communication graph is a ring, and the heterogeneous Lipschitz constants for 10 agents are given by 1.5, 3.7, 2.0, 1.5, 3, 3.3, 4.8, 6.5, 10.5 and 2.9, respectively. Subplot (b) represents the case when the communication graph is randomly generated using the Metropolis-Hastings weights selection rule for a random graph $G(10, 0.6)$ (see detailed explanation in 6.3.3), and the heterogeneous Lipschitz constants for 10 agents are given by 2.5, 1.7, 2.0, 1.5, 3, 3.3, 20, 6.5, 4 and 2.9, respectively. Subplot (c) represents the case when the communication graph is a star and the heterogeneous Lipschitz constants for 10 agents are given by 2.5, 1.7, 2.0, 1.5, 3, 3.3, 7.1, 6.5, 4 and 2.9, respectively.

**Handwritten digits classification on MNIST** In this experiment, we consider a neural network-based classifier for the MNIST dataset of handwritten digits (LeCun et al., 1994), a widely used benchmark for training and evaluating models in the field of machine learning (Deng, 2012). The dataset contains 60,000 training images and 10,000 testing images, with each set containing a roughly equal number of images for each digit from 0 to 9. We employ a deep convolutional neural network (CNN)–based classifier. The architecture begins with two convolutional layers, with 16 and 32 filters, respectively, each followed by a max pooling layer. Finally, the network includes a fully connected dense layer that maps the extracted features to 10 output classes. In the decentralized case, the data are partitioned across five agents in a label-based, heterogeneous manner under a fully connected graph. Specifically, all data entries with the same label are assigned to the same agent, and each agent holds data corresponding to two distinct labels. Each agent estimates the smoothness constant using 1,000 data entries. The results are summarized in Figure 7. It is evident that Algorithm 1 using local step sizes can substantially reduce the number of iterations required to reach an optimal solution.

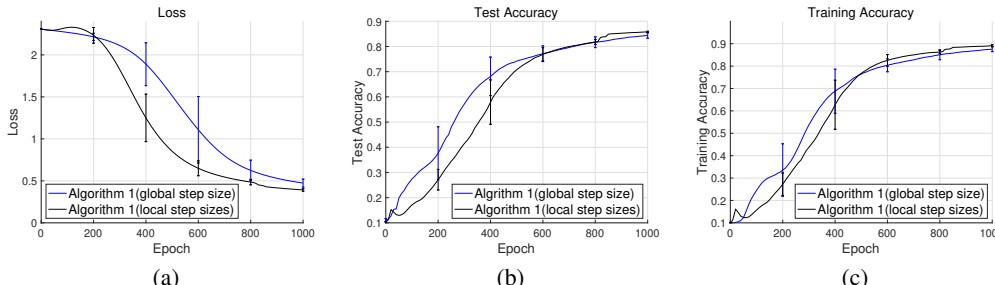

(a)        (b)        (c)

Figure 6: Comparison of Algorithm 1 using local step sizes versus a global step size on the MNIST dataset. The result shows that using local step sizes achieves faster convergence than using a global step size in terms of training loss, training accuracy, and test accuracy.

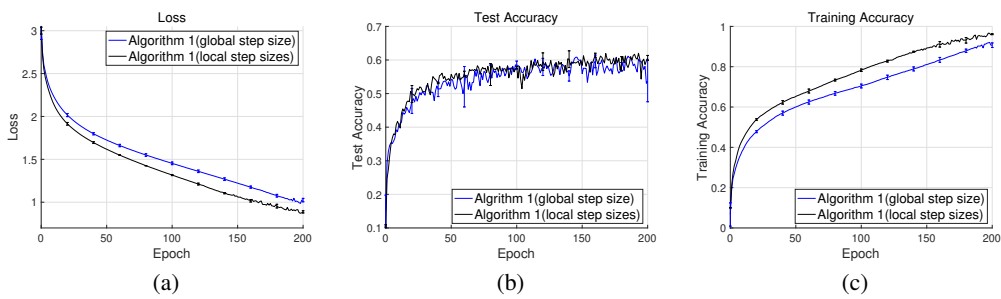

(a)        (b)        (c)

Figure 7: Comparison of Algorithm 1 using local step sizes versus a global step size on the CIFAR10 dataset. The result shows that using local step sizes achieves faster convergence than using a global step size in terms of training loss, training accuracy, and test accuracy.

**Image classification on CIFAR-10** In this experiment, we train a deep neural network on the CIFAR-10 dataset (Krizhevsky et al., 2009), which offers greater diversity and complexity compared to the MNIST dataset. The CIFAR-10 dataset consists of 60,000 color images of size 32×32 pixels, divided into 10 classes. Each class contains 6,000 images, with 5,000 allocated for training and 1,000 for testing. In this experiment, we employ a four-level CNN–based classifier. The architecture consists of four convolutional layers with 32, 64, 128, and 128 filters, respectively. Max pooling layers are applied after the second and fourth convolutional layers. Finally, the network includes a global average pooling layer and a fully connected dense layer that maps the extracted features to 10 output classes. In our implementation of the Algorithm 1, we partition the entire dataset across ten agents based on labels—that is, all data entries with the same label are assigned to the same agent on a fully connected graph. Similar to the previous experiment, each agent used a small number of samples to estimate its local smoothness constant. The results are summarized in Figure 6. It is evident that using local step sizes can substantially reduce the number of iterations required to reach an optimal solution.

**Remark 6.1.** The code for the additional learning-task experiments has been updated and is available in our anonymous GitHub repository, within the Neural_networks directory.

## 6.3 ADDITIONAL PEP RESULTS

### 6.3.1 RESULTS FOR RING TOPOLOGY

We also consider a non-all-to-all network, where four agents are connected through an undirected ring topology characterized by the following mixing matrix:

$$\begin{bmatrix} \frac{1}{3} & \frac{1}{3} & 0 & \frac{1}{3} \\ \frac{1}{3} & \frac{1}{3} & \frac{1}{3} & 0 \\ 0 & \frac{1}{3} & \frac{1}{3} & \frac{1}{3} \\ \frac{1}{3} & 0 & \frac{1}{3} & \frac{1}{3} \end{bmatrix}$$

The following figure shows the exact speed gain of Algorithm 1 over EXTRA and NIDS under this ring topology. A less-than-one ratio means that Algorithm 1 with local step sizes has a smaller optimization error than EXTRA and NIDS, and hence, there is a gain in convergence speed over EXTRA and NIDS.

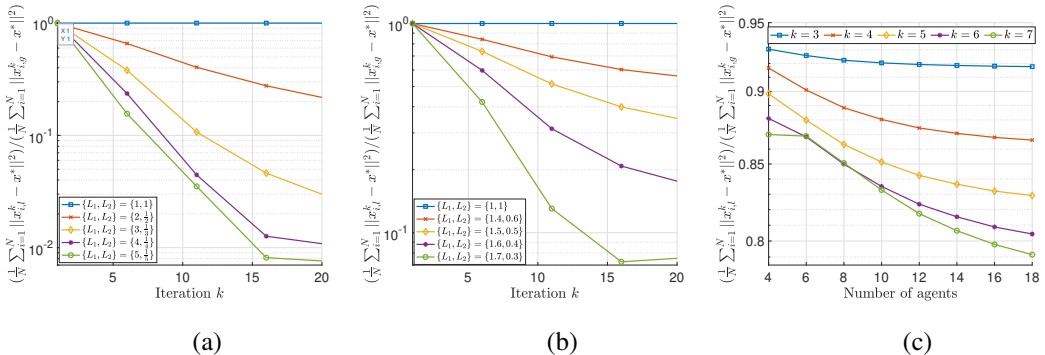

|       |       |       |
|-------|-------|-------|
| (a)   | (b)   | (c)   |

Figure 8: Convergence speed gain of Algorithm 1 with local step sizes over Algorithm 1 with global step sizes (which reduces to EXTRA) under a ring graph. The y-axis shows the ratio of optimization errors obtained by PEP under the local step size case versus the global step size case. A ratio less than 1 indicates superior convergence performance of the local step size strategy. (a) and (b) consider the four-agent ring topology under strong convexity with $\mu = 0.1$ and heterogeneous Lipschitz constants, where two agents have $L_1$ and the other two have $L_2$. In (a), the constants satisfy $L_1 \cdot L_2 = 1$, while in (b), they satisfy $L_1 + L_2 = 1$. (c) considers the more than four-agent setting with half of the agents having a Lipschitz constant of $\frac{1}{2}$, whereas the other half have a Lipschitz constant of 2. The strongly convex parameter is $\mu = 0.1$.

The four agents are divided into two groups, with each group assigned a Lipschitz smoothness constant of 2 and $\frac{1}{2}$, respectively. The following figure shows the performance of Algorithm 1 under different switch timings. As shown in the figure, Algorithm 1 consistently outperforms NIDS and EXTRA.

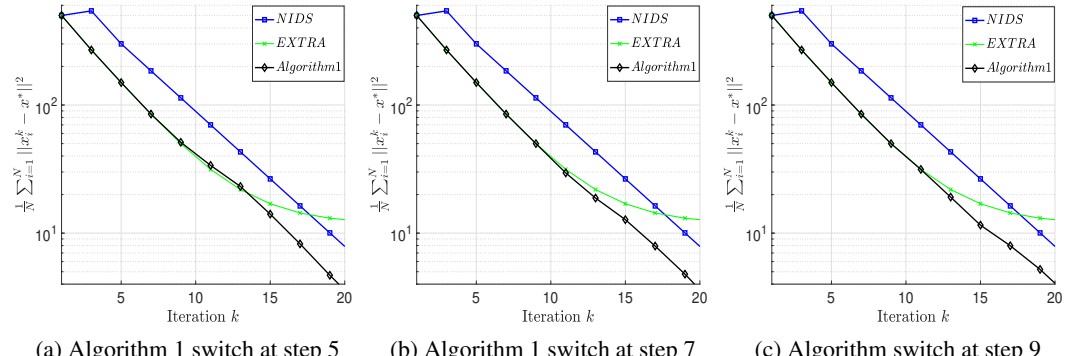

| (a) Algorithm 1 switch at step 5 | (b) Algorithm 1 switch at step 7 | (c) Algorithm switch at step 9 |
|----------------------------------|----------------------------------|--------------------------------|

Figure 9: Performance comparison of EXTRA, NIDS, and Algorithm 1 using PEP under a ring graph. The performance metric is the optimization error $\frac{1}{N}\sum_{i=1}^{N}||x_i^k - x^*||^2$. The switching time instant for Algorithm 1 was set to 5, 7, and 9, respectively, in subplots (a), (b), and (c). The blue curve represents NIDS with a global step size, which has the same update dynamics as Algorithm 1 under a global step size. The black curve represents the performance of Algorithm 1 with local step sizes. The green curve represents the performance of the EXTRA algorithm with local step sizes. In the evaluation, we consider a four-agent ring topology setting under strong convexity with parameter $\mu = 0.1$ and heterogeneous Lipschitz constants for two groups with $L_1 = \frac{1}{2}$ and $L_2 = 2$.

### 6.3.2 RESULTS FOR STAR TOPOLOGY

We also consider a network where four agents are connected through an undirected star topology characterized by the following mixing matrix:

$$\begin{bmatrix} \frac{1}{4} & \frac{1}{4} & \frac{1}{4} & \frac{1}{4} \\ \frac{1}{4} & \frac{3}{4} & 0 & 0 \\ \frac{1}{4} & 0 & \frac{3}{4} & 0 \\ \frac{1}{4} & 0 & 0 & \frac{3}{4} \end{bmatrix}$$

The following figure shows the exact speed gain of Algorithm 1 over EXTRA and NIDS. A less-than-one ratio means that Algorithm 1 with local step sizes has a smaller optimization error than EXTRA and NIDS, and hence, there is a gain in convergence speed over EXTRA and NIDS.

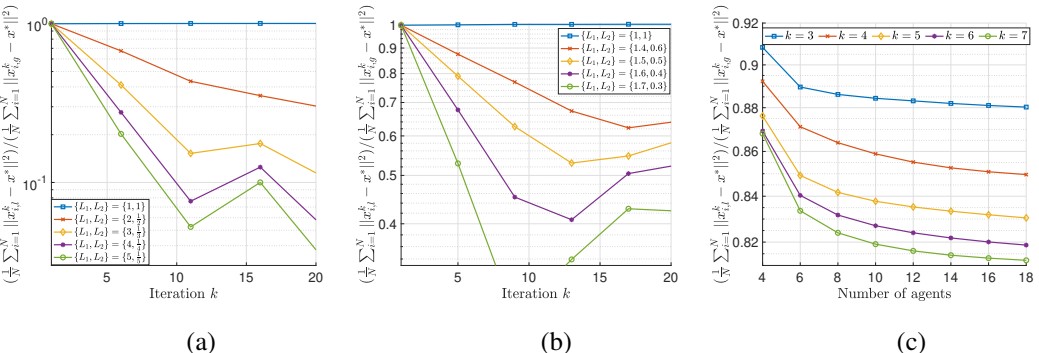

(a)                          (b)                          (c)

Figure 10: Convergence speed gain of Algorithm 1 with local step sizes versus Algorithm 1 with global step sizes (which reduces to EXTRA) under a star graph. The y-axis shows the ratio of optimization errors obtained by PEP under the local step size case versus the global step size case. A ratio less than 1 indicates superior convergence performance of the local step size strategy. (a) and (b) consider the four-agent ring topology under strong convexity with $\mu = 0.1$ and heterogeneous Lipschitz constants, where the central agent has a Lipschitz constant $L_1$ and the other agents have a common Lipschitz constant $L_2$. In (a), the Lipschitz constants satisfy $L_1 \cdot L_2 = 1$, while in (b), they satisfy $L_1 + L_2 = 1$. (c) considers the more than four-agent setting with three of the agents having a Lipschitz constant of $\frac{1}{2}$, whereas the other one has a Lipschitz constant of 2. The strongly convex parameter is always $\mu = 0.1$.

The following figure shows the performance of Algorithm 1 under different switch timings. As shown in this figure, Algorithm 1 consistently outperforms NIDS and EXTRA.

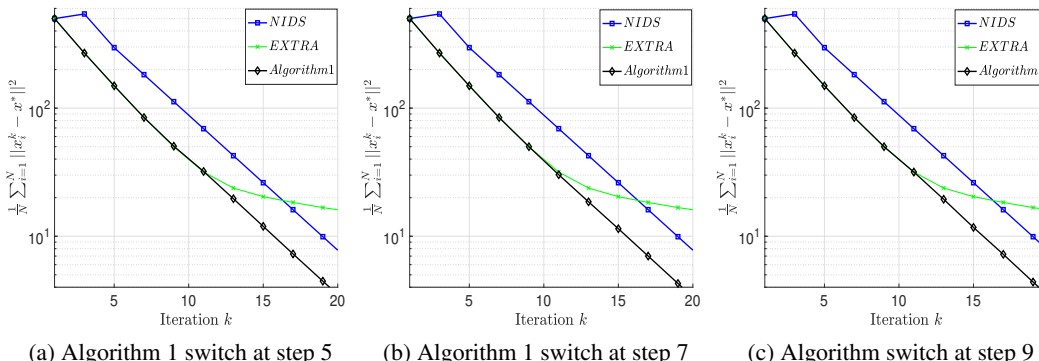

(a) Algorithm 1 switch at step 5      (b) Algorithm 1 switch at step 7      (c) Algorithm switch at step 9

Figure 11: Performance comparison of EXTRA, NIDS, and Algorithm 1 using PEP under a star graph. The performance metric is the optimization error $\frac{1}{N}\sum_{i=1}^{N}||x_i^k - x^*||^2$. The switching time instant for Algorithm 1 was set to 5, 7, and 9, respectively, in subplots (a), (b), and (c). The blue curve represents NIDS with a global step size, which has the same update dynamics as Algorithm 1 under a global step size. The black curve represents the performance of Algorithm 1 with local step sizes. The green curve represents the performance of the EXTRA algorithm with local step sizes. In the evaluation, we consider a four-agent star topology under strong convexity with parameter $\mu = 0.1$. The problem features heterogeneous Lipschitz constants: the central agent has a Lipschitz constant $L_1 = 2$, while the remaining agents have a common Lipschitz constant $L_2 = \frac{1}{2}$.

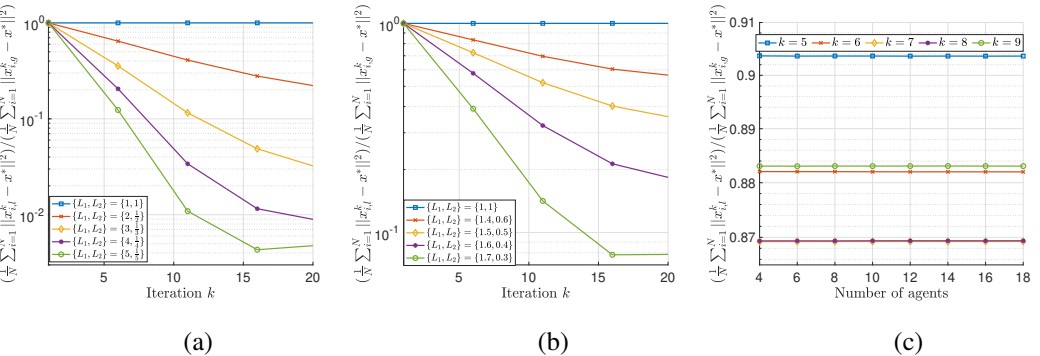

(a)               (b)               (c)

Figure 12: Convergence speed gain of Algorithm 1 with local step sizes versus Algorithm 1 with a global step size (which reduces to EXTRA) under an Erdős–Rényi random graph with $N = 4$ and $p = 0.6$. The y-axis shows the ratio of optimization errors obtained by PEP under the local step size case versus the global step size case. A ratio less than 1 indicates superior convergence performance of the local step size strategy. (a) and (b) consider the four-agent setting with $\mu = 0.1$ and heterogeneous Lipschitz constants, where two agents have Lipschitz constant $L_1$ and the other two have Lipschitz constant $L_2$. In (a), the Lipschitz constants satisfy $L_1 \cdot L_2 = 1$, while in (b), they satisfy $L_1 + L_2 = 1$. (c) considers the more than four-agent setting with $p = 0.6$, over 5 independent runs. In this setting, half of the agents have a Lipschitz constant of $\frac{1}{2}$, whereas the other half have a Lipschitz constant of 2. The strongly convex parameter is $\mu = 0.1$.

### 6.3.3 RESULTS FOR RANDOM TOPOLOGY

We also evaluated the performance of the proposed algorithm using an Erdős–Rényi random graph. The Erdős–Rényi graph model $G(N, p)$ generates a random undirected graph with $N$ nodes where each of the $\binom{N}{2}$ possible edges is included independently with probability $p \in [0, 1]$. We consider a random graph $G(N, p)$ with $p = 0.6$, and the mixing matrix is generated by using the Metropolis-Hastings weights selection rule. The Metropolis rule defines weights as:

$$
W_{ij} = \begin{cases} \frac{1}{1+\max(d_i, d_j)} & \text{if } i \neq j \text{ and } (i, j) \in E, \\ 1 - \sum_{k \in \mathcal{N}_i} W_{ik} & \text{if } i = j, \\ 0 & \text{otherwise,} \end{cases}
$$

where $(i, j) \in E$ indicates that there is an undirected edge between nodes $i$ and $j$, $d_i$ is the degree of node $i$, defined as the number of neighbors of node $i$, $\mathcal{N}_i$ is the set of neighbors of node $i$, i.e., $\mathcal{N}_i = \{j \mid (i, j) \in E\}$. Figure 12 shows the exact speed gain of Algorithm 1 over EXTRA and NIDS. A less-than-one ratio means that Algorithm 1 with local step sizes has a smaller optimization error than EXTRA and NIDS, and hence, there is a gain in convergence speed over EXTRA and NIDS.

The following figure shows the performance of Algorithm 1 under different switch timings. As shown in this figure, Algorithm 1 consistently outperforms NIDS and EXTRA.

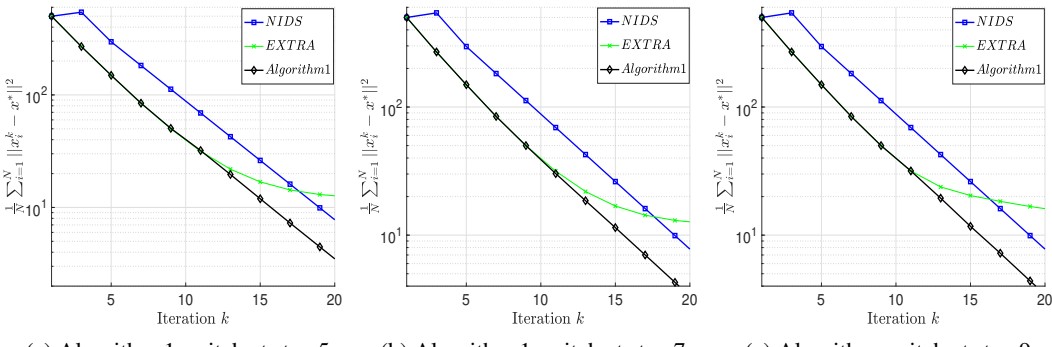

(a) Algorithm 1 switch at step 5    (b) Algorithm 1 switch at step 7    (c) Algorithm switch at step 9

Figure 13: Performance comparison of EXTRA, NIDS, and Algorithm 1 using PEP under a random graph with $N = 4$ and $p = 0.6$. The performance metric is the optimization error $\frac{1}{N} \sum_{i=1}^{N} \|x_i^k - x^*\|^2$. The switching time instant for Algorithm 1 was set to 5, 7, and 9, respectively, in subplots (a), (b), and (c). The blue curve represents NIDS with a global step size, which has the same update dynamics as Algorithm 1 under a global step size. The black curve represents the performance of Algorithm 1 with local step sizes. The green curve represents the performance of the EXTRA algorithm with local step sizes. In the evaluation, we consider a four-agent random topology under strong convexity with parameter $\mu = 0.1$ and heterogeneous Lipschitz constants for two groups $L_1 = 2$ and $L_2 = \frac{1}{2}$.

### 6.3.4 PROOF OF THEOREM 3.2

*Proof.* Firstly, we define $M = c^{-1}(I - W)^{\dagger} - \Lambda$, where $(I - W)^{\dagger}$ denotes the pseudo-inverse of $I - W$. Since $c \Lambda^{1/2}(I - W)\Lambda^{1/2} \preceq I$, we have $\sum_{i=1}^{N} \sum_{j=1}^{N} M_{ij} (x_i^k - x_i^*)^{\top}(x_j^k - x_j^*) \geq 0$ (see Proposition 1 in (Li et al., 2019a)). The iterations performed prior to the switching step produce a finite sequence. Let $k_s < \infty$ denote the final iteration before the switch, determined by $\mathcal{S}$. We assume that $\frac{1}{N} \sum_{i=1}^{N} \frac{1}{\alpha_i}\|x_i^{k_s} - x^*\|^2 + \sum_{i=1}^{N} \sum_{j=1}^{N} M_{ij} (x_i^{k_s} - x_i^*)^{\top}(x_j^{k_s} - x_j^*) \leq$

$C_0 \left( \frac{1}{N} \sum_{i=1}^{N} \frac{1}{\alpha_i}\|x_i^0 - x^*\|^2 + \sum_{i=1}^{N} \sum_{j=1}^{N} M_{ij} (x_i^0 - x_i^*)^{\top}(x_j^0 - x_j^*) \right)$, for some sufficiently large

scalar $C_0$. For all $k \geq k_s$, the update rule of Algorithm 1 coincides exactly with that of NIDS. Consequently, for every $K \geq k_s$, we get

$$\frac{1}{N} \sum_{i=1}^{N} \frac{1}{\alpha_i}\|x_i^K - x^*\|^2 \leq \rho^{K-k_s} * (\frac{1}{N} \sum_{i=1}^{N} \frac{1}{\alpha_i}\|x_i^{k_s} - x^*\|^2 + \sum_{i=1}^{N} \sum_{j=1}^{N} M_{ij} (x_i^{k_s} - x_i^*)^T(x_j^{k_s} - x_j^*))$$

$$\leq \rho^K * \frac{C_0}{\rho^{k_s}}(\frac{1}{N} \sum_{i=1}^{N} \frac{1}{\alpha_i}\|x_i^0 - x^*\|^2 + \sum_{i=1}^{N} \sum_{j=1}^{N} M_{ij} (x_i^0 - x_i^*)^T(x_j^0 - x_j^*))$$

$$\leq \rho^K C(\mathcal{S}, \{x_i^0\}_{i \in [N]}, \{\alpha_i\}_{i \in [N]}),$$

where

$$\rho = \max\left(1 - \left(2 - \max_i(\alpha_i L_i)\right) \min_i(\mu_i \alpha_i),\ 1 - c \lambda_{\max}\left(\Lambda^{-1/2}(I - W)^{\dagger}\Lambda^{-1/2}\right)\right) \in (0, 1),$$

and

$$C(\mathcal{S}, \{x_i^0\}_{i \in [N]}, \{\alpha_i\}_{i \in [N]}) = \frac{C_0}{\rho^{k_s}}(\frac{1}{N} \sum_{i=1}^{N} \frac{1}{\alpha_i}\|x_i^0 - x^*\|^2 + \sum_{i=1}^{n} \sum_{j=1}^{n} M_{ij} (x_i^0 - x_i^*)^T(x_j^0 - x_j^*)$$

This completes the proof. $\qquad\square$

