# OpenReview forum: "Local Stepsizes Accelerate Distributed Optimization"
_ICLR.cc/2026/Conference — Submitted to ICLR 2026_

### Official Review · Reviewer_5Jyh · 2025-10-27

**Soundness:** 3
**Presentation:** 2
**Contribution:** 3
**Rating:** 4
**Confidence:** 4

**Summary:**

This paper proposed a novel algorithm, which combines EXTRA and NIDS. Then, This paper analyzed the performance of the proposed method with PEP and showed that the proposed method can enjoy the advantages of both EXTRA and NIDS, converging faster than EXTRA and NIDS.

**Strengths:**

* This paper proposed a novel algorithm, which combines NIDS and EXTRA and uses a different stepsize among nodes.
* The performance of the proposed method was analyzed using PEP, and this paper demonstrated that the proposed method can converge faster than NIDS and EXTRA.

**Weaknesses:**

* Using the different stepsizes among the nodes is interesting, but how can we tune the stepsize in practice, e.g., for deep learning tasks? This paper evaluated the proposed method with simple convex functions, in which the smoothness can be computed easily. However, when we use the proposed methods for more complicated tasks, it is very expensive to tune the stepsize so that each node can use a different stepsize.
*  The convergence rate of the proposed method is not analyzed in this paper.
* The constraint shown in Eq. (5), $\sum_{i} \nabla f_i (x^\star) = 0$, is too strong. The quantity of $\frac{1}{N} \sum_i \nabla f_i (x^\star)$ is often used to measure the data heterogeneity for the convex optimization, e.g., [1], but this PEP analysis simplifies the problem and assumes that $\frac{1}{N} \sum_i \nabla f_i (x^\star)=0$.
* This paper only compares the proposed methods with NIDS and EXTRA. Comparing them with other optimization methods, such as decentralized gradient descent and SONATA [2], would strengthen this paper.


## Reference
[1] Koloskova et. al.,  A unified theory of decentralized SGD with changing topology and local updates. In ICML 2020

[2]  Sun et. al., Distributed Optimization Based on Gradient-tracking Revisited: Enhancing Convergence Rate via Surrogation, SIAM 2020

**Questions:**

See the weakness section.

---

> ### Author Response · Authors · 2025-11-19
> **Response to Reviewer 5Jyh**
>
> **Response to Reviewer $\\rm\\color{purple}{5Jyh}$**:
>
> We sincerely thank you for the time and effort you devoted to reviewing our manuscript. Below, we provide clarifications for some concerns that appear to stem from misunderstandings. We have also made revisions to improve the paper. Please find our detailed responses below:
>
> ${\\color{blue} \\text{Response to the weakness on tuning local step size in practice}}$: We argue that our use of local step sizes does not significantly increase the overall cost of stepsize tuning compared with the conventional setting, where a single universal stepsize must be selected. Although each agent determines its own (time-invariant) stepsize, the tuning cost per agent is substantially lower than that of tuning a universal stepsize. This is because each agent uses only its local dataset—a small portion of the global dataset—to select its step size, thereby greatly reducing the computational effort compared to centralized tuning.
>
> Moreover, in distributed optimization problems—the setting considered in this paper—tuning a universal stepsize is considerably more challenging. The data are stored locally, and each agent can only communicate with its nearest neighbors; no central coordinator exists to facilitate global tuning based on all agents’ data. In contrast, allowing each agent to tune its own step size using its own local data avoids this difficulty and simplifies the overall step size selection process in distributed learning.
>
> In the revised version of the paper, we have also added deep-learning experiments on the MNIST and CIFAR-10 benchmark datasets in Sec. 6.2 of the Appendix (Figure 6 and Figure 7). The results confirm that local step sizes continue to accelerate distributed learning, and that determining a local step size using only local data requires approximately $1/N$ of the time needed to determine a global step size, where $N$ is the number of agents. This confirms that our approach can be effectively applied to more complex learning tasks.
>
> ${\\color{blue} \\text{Response to the weakness on the convergence rate of the proposed method}}$: In the initial version, we did not provide the convergence rate because, after the switching step, our algorithm reduces to NIDS and therefore inherits its convergence rate. Following the reviewer’s suggestion, we have added a new Theorem 3.2 on page 6 that explicitly states the convergence rate (reproduced below for the reviewer’s convenience).
>
> **Theorem 3.2**
> Consider the distributed optimization problem (1) where each $f\_i$ is $\\mu\_i$ strongly convex and $L\_i$ smooth, and the mixing matrix $W$ satisfies the conditions specified in Definition 2.1. Suppose the step sizes satisfy $\\alpha\_i \\le \\frac{2}{L\_i}$ for all $i \\in [N]$, and
> $c$ is chosen such that $I - c\\,\\Lambda^{1/2}(I - W)\\Lambda^{1/2} \\succeq 0$. Let $\\{x\_i^K\\}\_{i\\in[N]}$ be the iterates generated by Algorithm 1. Then for all $K \\ge 1$, we have$$\\frac{1}{N}\\sum\_{i=1}^N \\frac{1}{\\alpha\_i}\\,\\|x\_i^{K} - x^{\\ast}\\|^2\\;\\le\\;\\rho^{K}\\, C(\\mathcal{S}, \\{x\_i^0\\}\_{i\\in[N]}, \\{\\alpha\_i\\}\_{i\\in[N]}),$$
> where
> $$
> \\rho = \\max\\!\\left(
> 1 - \\left(2 - \\max\_{i\\in[N]} (\\alpha\_i L\_i)\\right)\\min\_{i\\in[N]} (\\mu\_i \\alpha\_i),\\;1 - c\\,\\lambda\_{\\max}\\!\\left(\\Lambda^{-1/2}(I - W)^{\\dagger} \\Lambda^{-1/2}\\right)\\right)\\in (0,1)
> $$
> and $C(\\mathcal{S}, \\{x\_i^0\\}\_{i\\in[N]}, \\{\\alpha\_i\\}\_{i\\in[N]})$ is a constant depending on the switching condition $\\mathcal{S}$, the initialization $\\{x\_i^0\\}$, and the step sizes $\\{\\alpha\_i\\}$.
>
> ${\\color{blue} \\text{Response to the question on Eq (5)}}$: We would like to argue that we use $x^{\\ast}$ to denote the global optimum and $x^{\\star}\_i$ to denote each local optimum, respectively. In Eq (5), the constraint $ \\nabla f(x^{\\ast}) = \\sum\_{i=1}^{N}\\nabla f\_i(x^{\\ast}) = 0$ makes sure that $x^{\\ast}$ is the global optima. We do **not** require $\\frac{1}{N}\\sum\_{i=1}^N\\nabla f\_i(x^{\\star})$ on local optima.
>
> ${\\color{blue} \\text{Response to the question on comparing with more algorithms}}$: Thanks for the comment. We have included comparisons with DGD [2] and SONATA [3], as shown in the updated Figures 3 and 4. The step sizes for DGD and SONATA were fine-tuned to ensure their best possible performance, yet our algorithm still achieves superior results. It is worth noting that our goal is not to claim speedups over all existing methods. Rather, we aim to show that significant improvements in convergence speed can be achieved through a simple yet impactful design: employing different step sizes at different stages of the iteration process.
>
> [1]Nedic, Angelia and Ozdaglar, Asuman. Distributed Subgradient Methods for Multi-Agent Optimization, IEEE Trans. Autom. Control, 2009.
>
> [2]Sun et. al., Distributed Optimization Based on Gradient-tracking Revisited: Enhancing Convergence Rate via Surrogation, SIAM J. Optim., 2020.

---

### Official Review · Reviewer_ZQLq · 2025-11-01

**Soundness:** 4
**Presentation:** 4
**Contribution:** 3
**Rating:** 6
**Confidence:** 4

**Summary:**

This paper presents a rigorous and systematic theoretical discovery that uncoordinated stepsize in distributed optimization could lead to faster convergence speed. This work builds on the powerful performance Estimation Problem framework and converts convergence analysis into an SDP formulation. The authors then show for EXTRA and NIDS type of algorithm, both worst case and average case performance can improve under the right stepsize choices.

**Strengths:**

I really enjoyed the theoretical framework to analyze performance of distributed optimization methods. I believe the research community's view is not that we prefer coordinated stepsize, we would rather have uncoodinated stepsize for implementation purposes, but it was very hard to guarantee convergence with those. This paper precisely handles that scenario and gives answers for when we could get convergence guarantees. The use of SDP and interpolation enables stability/convergence analysis of various methods in a systematic way. Numerical results compared both average case and worst case performances.

**Weaknesses:**

Previously stepsize $1/L_i$ was analyzed and failed to converge for some problems. I believe the fix here is actually Eq (7). The authors assumed that individual optimal solutions are close to the global solution and therefore even if the methods pull towards the local minima, the final solution won't be too far away from the true solution. This is a somewhat cheating assumption. Sure for any problem realization, such R exists, but the reason for divergence is precisely the fact that R is not equal to 0. So rather than saying the different stepsizes are helpful, it was really a small R that enabled good performance. When R is small, almost all distributed optimization method will work beautifully, as this separation of local min was the reason behind zig-zag in many previous papers.

**Questions:**

Can the authors propose the "best" stepsize rule of thumb? Beyond having a structured analysis of performance, what does this PEP framework give us? Can this be done in a distributed way?

---

> ### Author Response · Authors · 2025-11-19
> **Response to reviewer ZQLq**
>
> **Response to Reviewer $\\rm\\color{purple}{ZQLq}$**:
>
> We appreciate your time and comments. Please see our detailed response below:
>
> ${\\color{blue} \\text{Response to Weakness on the use of Eq(7)}}$: In all of our PEP settings (including Eq. (7)), we do **not** assume that the individual optimal solutions are close to the global optimum. In Eq. (7), $R_*$ is only introduced to ensure that the distance between each individual optimal solution and the global optimum is bounded; it is never assumed to be very small or close to zero. In fact, in the comparison between Algorithm 1 and NIDS/EXTRA in Figure 1—where the same $R_*$ is used for all algorithms—EXTRA (the green curve) with step sizes $\\frac{1}{L\_i}$ fails to converge, indicating that our used $R_*$ is not small. Furthermore, as stated in the proof of Theorem 3.1 in Appendix 6.1, our results rely solely on the assumption that every local optimum lies no farther from the global optimum than the initial point—that is, the distance from any local optimum to the global optimum does not exceed the distance from the initial point to the global optimum ($R_* \leq R_0$).
>
> ${\\color{blue} \\text{Response to question on the selection of ``best" step size and what PEP gives us}}$:
>
> Our goal is not to determine the best step size. Instead, our goal is to rigorously establish that local step sizes can lead to faster convergence in distributed optimization for general smooth and strongly convex objective functions. We also introduce a new distributed optimization algorithm that effectively exploits local step sizes to accelerate convergence. To make this clear, in the revised manuscript, we have formalized these results as Theorem 3.2 (page 6) and Theorem 3.3 (page 8).
>
> We would like to note that PEP can also be used to optimize the step size by treating it as a decision variable and solving the corresponding performance estimation problem. This approach has been demonstrated in [1] for centralized gradient descent, and has been used in other work to tune step sizes and other algorithmic parameters [2]. However, this optimization problem is very challenging, even for simple centralized gradient methods, as it is typically non-convex and can be computationally expensive, as noted in [1]. While we cannot provide a definitive answer at this stage on whether an optimal step size strategy can be determined for distributed optimization, we consider this a promising direction for future work.
>
>
> ${\\color{blue} \\text{Response to question on if this can be done in a distributed way}}$: With respect to identifying an optimal step size via PEP, we believe that this task cannot be addressed in a purely distributed manner, as the interdependence among neighboring agents exerts nontrivial influences on global convergence. Thus, any attempt to employ PEP for determining an optimal step  size would necessarily require formulating the distributed optimization algorithm as a single, centralized PEP problem that captures all agent interactions simultaneously.
>
> However, for our proposed strategy, which does not require optimizing the step size, the step size selection can be carried out naturally in a distributed manner. This is because our algorithm uses local step sizes that are independent of the network topology, and each step size is determined solely by its corresponding local loss function.
>
> Additionally, we have added Theorem 3.2 on page 6 of the revised manuscript to provide the analytical convergence rate of Algorithm 1. We have also compared Algorithm 1 with other distributed optimization algorithms, such as DGD [3] and SONATA [4], as shown in Figures 3 and 4, where the step sizes for these baselines are hand-tuned to achieve their best observed performance. Furthermore, we have included additional experiments on regression and deep-learning tasks on benchmark datasets MNIST and CIFAR-10, presented in Appendix 6.2 (Figures 5–7), to further demonstrate the efficiency and effectiveness of our algorithm.
>
> [1]Das Gupta et. al., Branch-and-bound performance estimation programming: A unified methodology for constructing optimal optimization methods, Mathematical Programming, 2024.
>
> [2]Adrien Taylor and Yoel Drori., An optimal gradient method for smooth strongly convex minimization, arXiv2101.09741.
>
> [3]Nedic, Angelia and Ozdaglar, Asuman., Distributed Subgradient Methods for Multi-Agent Optimization, IEEE Transactions on Automatic Control, 2009.
>
> [4]Sun et. al., Distributed Optimization Based on Gradient-tracking Revisited: Enhancing Convergence Rate via Surrogation, SIAM Journal on Optimization, 2020.

---

### Official Review · Reviewer_M7FA · 2025-11-01

**Soundness:** 1
**Presentation:** 3
**Contribution:** 1
**Rating:** 2
**Confidence:** 5

**Summary:**

This paper explores the impact of local step sizes on the convergence of decentralized optimization. The paper states that local step sizes can lead to faster convergence in distributed optimization compared to a global step size, although this paper does not provide any theoretical guarantees for this claim. Additionally, the paper introduces a new algorithm also without any theoretical guarantees, which may be faster in some settings compared to other algorithms.

This paper should be rejected for the following reasons: (1) the results are not well justified by either theory or practice, (2) the experiments are conducted with insufficient set parameters, and (3) The new algorithm (Algorithm 1), provided without theoretical guarantees of convergence.

**Strengths:**

This paper introduces the interesting research question: “Can local step sizes respecting local geometries of individual objective functions outperform their counterpart with a global universal step size in distributed optimization under general strongly convex objective functions?”

**Weaknesses:**

**Main argument**

To prove the rejection, I provide the following contradictions for each of the contributions:

The first stated contribution of this paper is "rigorously proving that local step sizes can indeed provide faster convergence in distributed optimization than their global step sizes". However, this paper does not include any rigorous theoretical guarantees for this claim. Can you provide proof for this statement?

 The second contribution claimed by the authors is "we have also developed a new distributed optimization algorithm that can exploit local step sizes to accelerate distributed optimization", but the theoretical guarantees for the algorithm are not provided in the paper. Can you provide proof or any kind of evidence that the proposed approach is faster than others?

 The third contribution stated in this paper is “ To rigorously compare the performance of distributed optimization using local step sizes versus global coordinated step sizes, we generalized the existing distributed PEP framework in two key aspects: firstly, we modified it to accommodate the boundedness restriction of optimal solutions in practical applications of distributed optimization; secondly, we revised it to reduce computational complexity, which is crucial given that the existing PEP approach is computationally intensive.”.  However, using only the PEP framework is not sufficient for a rigorous comparison between the use of local and global step sizes in distributed optimization. Usually, authors are inspired by the use of PEP frameworks to formulate rigorous theoretical results, for example, arXiv:1803.06600, arXiv:2101.09741.

The fourth contribution stated as “we conduct a comprehensive set of experiments on both synthetic and real-world datasets to validate the correctness of our theoretical findings and demonstrate the effectiveness of the proposed algorithm.”.  But these experiments are not sufficient or complete due to the fact that they only use two functions or groups of functions. Also, the smoothness constants L_i differ by no more than a factor of 10. The experiments do not provide convincing evidence of the correctness of the key statement.

**Questions:**

**Things to improve the paper that did not impact the score:**

1. Provide theoretical guarantees for each statements in this work

2. Provide experiments for different settings of parameters

---

> ### Author Response · Authors · 2025-11-19
> **Response to Reviewer M7FA---Part I**
>
> **Response to Reviewer $\\rm\\color{purple}{M7FA}$**:
>
> We sincerely thank you for the time and effort you devoted to reviewing our manuscript. In response to your comments, we have made several changes to improve the manuscript. First, we formalized our theoretical findings in Theorem 3.3 on page 8 of the revised manuscript. Second, we conducted additional regression and machine-learning experiments with more parameter settings, as shown in Appendix 6.2. Third, we provide Theorem 3.2 on page 6, which gives the convergence guarantee for the proposed Algorithm 1. Detailed responses addressing each point are provided below.
>
> ${\\color{blue} \\text{Response to question on first contribution on local step size for better performance}}$: We emphasize that the PEP-based results are formally valid and mathematically rigorous, as they provide and compare the **exact** worst-case bounds for optimization algorithms over a general class of functions. In the revised manuscript, we have formalized these results as Theorem 3.3 on page 8 (reproduced below for the reviewer’s convenience).
>
> **Theorem 3.3**
> Consider the distributed optimization problem defined in (1), where the global function $f$ belongs to the class $\\mathcal{F}\_{\\mu, L}$, characterized by the smoothness parameter $L$ and strong convexity parameter $\\mu$. Let $\\mathcal{A}\_l$ denote Algorithm 1 with local step sizes $\\frac{1}{L\_i}$, and $\\mathcal{A}\_g$ denote Algorithm 1 with a global step size $\\frac{1}{\\bar{L}}$. For the performance metric
> $$
> P\_{\\mathcal{A}} \\coloneqq
> \\max\_{f\\in\\mathcal{F}\_{\\mu,L}}
> \\{
> \\frac{1}{N}\\sum\_{i=1}^N ||x\_{i,\\mathcal{A}}^{k} - x^\\ast||^2
> \\},
> $$
> which measures the worst-case optimization error at iteration $k$ over all $f\\in\\mathcal{F}\_{\\mu,L}$. Consider the ratio $P\_{\\mathcal{A}\_l}/P\_{\\mathcal{A}\_g}$ as the relative acceleration of the local step sizes method compared with the global step size method (with values below $1$ indicating acceleration and smaller ratios indicating stronger acceleration). Figure 2 gives the exact ratio values under different function classes, and the ratio values being strictly less than 1 for all tested function classes $\\mathcal{F}\_{\\mu, L}$ in the heterogeneous setting confirm that Algorithm 1 with local step sizes achieves a strictly faster convergence rate than its global step size counterpart under all considered function classes.
>
> Note that, leveraging PEP, we compare the performance of algorithms using the **exact** optimization error, in contrast to traditional comparisons typically reling on **loose** upper bounds that can be overly conservative and sometimes misleading [1]. Because our PEP based approach produces an exact bound, it identifies a specific problem instance on which Algorithm 1 performs exactly as poorly as the bound indicates. With this interpretation, the speed gain ratio $P\_{\\mathcal{A}\_l}/P\_{\\mathcal{A}\_g}$ directly compares the exact worst-case optimization errors of Algorithm 1 when using local step sizes versus using a global step size. A speed gain ratio less than one therefore means that—in the worst case—using local step sizes leads to a strictly smaller optimization error at iteration $k$. Hence, this ratio quantitatively captures the exact speed gain achieved by adopting local step sizes in Algorithm 1. **To the best of our knowledge, such a conclusion cannot be obtained from any conventional conservative analytical analyses**.
>
> [1]Meunier, et al., Several Performance Bounds on Decentralized Online Optimization are Highly Conservative and Potentially Misleading, arXiv:2509.06466.
>
> (Continued on Part---II)

---

> ### Author Response · Authors · 2025-11-19
> **Response to Reviewer M7FA---Part II**
>
> ${\\color{blue} \\text{Response to the question on the second contribution of theoretical guarantee of the proposed algorithm}}$: In the initial version, we did not provide a theorem for the convergence of the algorithm because, after the switching step, our algorithm reduces to NIDS and therefore inherits the convergence guarantees established for NIDS. In the revised version, following the reviewer’s suggestion, we have clarified this point by adding a new Theorem 3.2 on page 6 that explicitly states the theoretical guarantee for the proposed algorithm (reproduced below for the reviewer’s convenience).
>
> **Theorem 3.2**
> Consider the distributed optimization problem (1) where each $f\_i$ is $\\mu\_i$ strongly convex and $L\_i$ smooth, and the mixing matrix $W$ satisfies the conditions specified in Definition 2.1. Suppose the step sizes satisfy $\\alpha\_i \\le \\frac{2}{L\_i}$ for all $i \\in [N]$, and
> $c$ is chosen such that $I - c\\,\\Lambda^{1/2}(I - W)\\Lambda^{1/2} \\succeq 0$. Let $\\{x\_i^K\\}\_{i\\in[N]}$ be the iterates generated by Algorithm 1. Then for all $K \\ge 1$, we have$$\\frac{1}{N}\\sum\_{i=1}^N \\frac{1}{\\alpha\_i}\\,\\|x\_i^{K} - x^{\\ast}\\|^2\\;\\le\\;\\rho^{K}\\, C(\\mathcal{S}, \\{x\_i^0\\}\_{i\\in[N]}, \\{\\alpha\_i\\}\_{i\\in[N]}),$$
> where
> $$
> \\rho = \\max\\!\\left(
> 1 - \\left(2 - \\max\_{i\\in[N]} (\\alpha\_i L\_i)\\right)\\min\_{i\\in[N]} (\\mu\_i \\alpha\_i),\\;1 - c\\,\\lambda\_{\\max}\\!\\left(\\Lambda^{-1/2}(I - W)^{\\dagger} \\Lambda^{-1/2}\\right)\\right)\\in (0,1)
> $$
> and $C(\\mathcal{S}, \\{x\_i^0\\}\_{i\\in[N]}, \\{\\alpha\_i\\}\_{i\\in[N]})$ is a constant depending on the switching condition $\\mathcal{S}$, the initialization $\\{x\_i^0\\}$, and the step sizes $\\{\\alpha\_i\\}$.
>
> Note that we do not make any comparisons using the bound in this theorem, as this analytically established bound is loose (as is the case for most analytical results in general distributed optimization problems), and using it for comparison would be misleading or fundamentally meaningless [1]. The efficiency of our algorithm is fully supported by the exact speed-gain ratios in Figure 2, as formalized in Theorem 3.3. It is also important to emphasize that our goal is not to claim universal speedups over all existing methods. Rather, our aim is to rigorously establish the advantages of local step sizes over a global step size for general smooth and strongly convex objective functions, through a simple yet impactful design illustrated in Algorithm 1. Additional theoretical results for sparse graphs are provided in Appendix 6.3.
>
> ${\\color{blue} \\text{Response to the question on the third contribution}}$: We argue that PEP is sufficient for performance comparison in our paper, as formalized in the newly added Theorem 3.3. In fact, we follow a similar approach to that in the mentioned paper arXiv:1803.06600, where PEP is used to select step-size coefficients that reduce the gradient norm for smooth convex functions. The second mentioned reference (arXiv:2101.09741) adopts a closely related strategy as well. In both references, the efficiency of an algorithm is evaluated using the same criterion as in our paper: the algorithm returning smaller worst-case optimization error under PEP has better performance. In fact, as shown in Theorem 3.3 of our revised manuscript, a speed-gain ratio less than one, such as those reported in Figure 2, directly indicates a speedup of the same algorithm achieved by adopting local step sizes. The only distinction between our work and the studies in the two mentioned papers is that, while those works use PEP to guide algorithm design, we instead use the exact optimization error from PEP to directly compare the performance of different algorithms. Furthermore, our observed speedup is not achieved through a highly structured or intricate step-size selection scheme as these in arXiv:1803.06600 and arXiv:2101.09741, but rather through a simple and effective strategy of employing different step sizes at different stages.
>
>
> [1]Meunier, et al., Several Performance Bounds on Decentralized Online Optimization are Highly Conservative and Potentially Misleading, arXiv:2509.06466.
>
> (Continued on Part---III)

---

> ### Author Response · Authors · 2025-11-20
> **Response to Reviewer M7FA---Part III**
>
> ${\\color{blue} \\text{Response to the question on experiments}}$: Following the suggestion, we further conducted additional experiments on a broader set of function classes under sparse communication graphs and deep-learning tasks, as summarized in Appendix 6.2 (Figures 5, 6, and 7). In our experimental setup, we increased the heterogeneity of the smoothness constants by more than a factor of 10 when using synthetic data. For the newly added deep-learning tasks on benchmark datasets MNIST and CIFAR-10, the smoothness constants are estimated based on data samples for each local loss function. The results demonstrate that adopting local step sizes can lead to faster convergence compared with using a universal global step size. Additionally, we include comparisons with distributed optimization algorithms such as DGD [2] and SONATA [3], as shown in the updated Figures 3 and 4. The step sizes for DGD [2] and SONATA [3] were carefully fine-tuned to ensure their best possible performance, yet our algorithm still achieves superior results.
>
> [2]Nedic et. al., Distributed Subgradient Methods for Multi-Agent Optimization, IEEE Transactions on Automatic Control, 2009.
>
> [3]Sun et. al., Distributed Optimization Based on Gradient-tracking Revisited: Enhancing Convergence Rate via Surrogation, SIAM Journal on Optimization, 2020.

---

### Author Response · Authors · 2025-11-27

Dear Reviewer,

I hope everything is going smoothly. I am writing to kindly ask whether it might be possible for you to provide your follow-up comments on our rebuttal. We believe we have addressed the concerns you raised, and we would greatly appreciate hearing your updated perspectives.

Thank you very much for your time and effort.

Sincerely,
The Authors

---

### Meta-Review · Area_Chair_drEp · 2026-01-07

**Summary:**

The paper proposed utilizing local step sizes to accelerate distributed optimization and introduced a hybrid algorithm combining EXTRA and NIDS dynamics, supported by a Performance Estimation Problem (PEP) analysis. While some concerns are addressed in the rebuttal and revised paper, such as the inclusion of more comprehensive experiments with higher heterogeneity and deep learning benchmarks, some key concerns are not fully addressed, including the lack of a rigorous, generalized theoretical proof for the claimed acceleration. The provided "theoretical guarantee" (Theorem 3.3) relies on numerical observations of specific instances (Figure 2) rather than a symbolic derivation valid for the entire class of smooth and strongly convex functions

**Reviewer Concerns:**

### [M7FA]

Weakness 1: Not fully addressed. The added Theorem 3.3 does not constitute a rigorous theoretical proof in the conventional sense. Instead of providing a symbolic derivation showing that the convergence rate of local step sizes is strictly superior to global step sizes for all valid problem instances, the theorem relies entirely on numerical verification (PEP/SDP results) presented in Figure 2. Since Figure 2 is limited to specific configurations (e.g., fixed $\mu=0.1$ and specific relationships between $L_i$), this amounts to a "proof by example" rather than a generalized proof for the entire parameter space of distributed optimization problems.

Weakness 2: Not fully addressed. The added Theorem 3.2 establishes linear convergence of Algorithm 1, but this guarantee is essentially inherited from NIDS because, after the switching step, the method follows NIDS updates. As a result, Theorem 3.2 does not substantiate the claimed acceleration relative to existing algorithms; it mainly certifies convergence. The rebuttal then relies on Theorem 3.3/Figure 2 for quantitative speed gains, which brings us back to Weakness 1: those gains are certified only for specific instantiated settings via PEP, rather than being proved uniformly over broad classes of $\{L_i\}$ and graphs.

Weakness 3: Not addressed. The rebuttal misunderstands the reviewer’s concern regarding "rigorous comparison." While the authors argue that PEP provides exact numerical bounds (unlike loose analytical bounds), the reviewer’s point is that running a numerical SDP for specific parameters constitutes empirical verification, not a generalized theoretical proof. The cited references (e.g., arXiv:1803.06600) typically use PEP as a tool to aid in formulating generalizable analytical results or designing optimal parameters. By contrast, this work stops at the numerical output. Since "Theorem 3.3" serves as the foundation for this comparison and is limited to the specific instances plotted in Figure 2 (as noted in Weakness 1), the authors have not provided a rigorous mathematical comparison that holds for the general class of functions they claim to address.

Weakness 4 is addressed.

### [ZQLq]

All concerns are addressed.

### [5Jyh]

Weakness 1: Not fully addressed. The rebuttal argues local stepsizes reduce tuning cost because each agent tunes on smaller data (cheaper per trial), but it does not address the trial-and-error nature of LR tuning in many tasks. In a coupled distributed method, a single overly large local stepsize can destabilize training, so the probability of a failed run may increase with the number of agents.

Other concerns are addressed.

**Reviewer Scores:**

The AC anticipates the following outcomes:

Reviewer M7FA will likely retain their score of 2, as their primary concerns remain unaddressed.

Reviewer ZQLq is expected to maintain a score of 6; although their issues have been resolved, the mixed feedback from the other reviewers effectively caps their score.

Reviewer 5Jyh may either maintain a 4 or upgrade to a 6, given that the majority of their concerns have been addressed.

---

### Decision · Program_Chairs · 2026-01-26

Reject